# Human DECR1 is an androgen-repressed survival factor that regulates PUFA oxidation to protect prostate tumor cells from ferroptosis

Zeyad D Nassar[1,2†], Chui Yan Mah[1,2†], Jonas Dehairs[3], Ingrid JG Burvenich[4], Swati Irani[1,2], Margaret M Centenera[1,2], Madison Helm[1,2], Raj K Shrestha[5], Max Moldovan[2], Anthony S Don[6], Jeff Holst[7], Andrew M Scott[4], Lisa G Horvath[8], David J Lynn[2,9], Luke A Selth[1,5,9], Andrew J Hoy[10], Johannes V Swinnen[3], Lisa M Butler[1,2]*

[1]University of Adelaide Medical School and Freemasons Foundation Centre for Men's Health, University of Adelaide, Adelaide, Australia; [2]South Australian Health and Medical Research Institute, Adelaide, Australia; [3]KU Leuven- University of Leuven, LKI- Leuven Cancer Institute, Department of Oncology, Laboratory of Lipid Metabolism and Cancer, Leuven, Belgium; [4]Tumour Targeting Laboratory, Olivia Newton-John Cancer Research Institute, and School of Cancer Medicine, La Trobe University, Melbourne, Australia; [5]Dame Roma Mitchell Cancer Research Laboratories, University of Adelaide, Adelaide, Australia; [6]NHMRC Clinical Trials Centre, and Centenary Institute, The University of Sydney, Camperdown, Australia; [7]Translational Cancer Metabolism Laboratory, School of Medical Sciences and Prince of Wales Clinical School, UNSW Sydney, Sydney, Australia; [8]Garvan Institute of Medical Research, NSW 2010; University of Sydney, NSW 2006; and University of New South Wales, Darlinghurst, Australia; [9]College of Medicine and Public Health, Flinders University, Bedford Park, Australia; [10]Discipline of Physiology, School of Medical Sciences, Charles Perkins Centre, Faculty of Medicine and Health, The University of Sydney, Camperdown, Australia

*For correspondence:
lisa.butler@adelaide.edu.au

†These authors contributed equally to this work

Competing interests: The authors declare that no competing interests exist.

**Abstract** Fatty acid β-oxidation (FAO) is the main bioenergetic pathway in human prostate cancer (PCa) and a promising novel therapeutic vulnerability. Here we demonstrate therapeutic efficacy of targeting FAO in clinical prostate tumors cultured ex vivo, and identify *DECR1*, encoding the rate-limiting enzyme for oxidation of polyunsaturated fatty acids (PUFAs), as robustly overexpressed in PCa tissues and associated with shorter relapse-free survival. *DECR1* is a negatively-regulated androgen receptor (AR) target gene and, therefore, may promote PCa cell survival and resistance to AR targeting therapeutics. DECR1 knockdown selectively inhibited β-oxidation of PUFAs, inhibited proliferation and migration of PCa cells, including treatment resistant lines, and suppressed tumor cell proliferation and metastasis in mouse xenograft models. Mechanistically, targeting of DECR1 caused cellular accumulation of PUFAs, enhanced mitochondrial oxidative stress and lipid peroxidation, and induced ferroptosis. These findings implicate PUFA oxidation via DECR1 as an unexplored facet of FAO that promotes survival of PCa cells.

## Introduction

Prostate cancer (PCa) is the most prevalent male cancer and the second leading cause of cancer deaths in men in Western societies (*Bray et al., 2018*). For patients with locally-recurrent and/or metastatic disease, androgen deprivation therapy (ADT) has remained the frontline strategy for clinical management since the 1940s (*Huggins and Hodges, 1941*), due to the dependence of PCa cells on androgens for growth and survival. Although ADT is initially effective in most patients, ultimately all will relapse with castration resistant prostate cancer (CRPC), which remains incurable. The failure of ADT is attributed to the emergence of adaptive survival pathways that reprogram androgen signaling and/or activate alternative tumor survival pathways. Consequently, the development and FDA approval of agents that more effectively target androgen signaling, including enzalutamide (ENZ, Xtandi; an AR antagonist) (*Tran et al., 2009*; *Cai and Balk, 2011*; *Rodrigues et al., 2014*), has expanded the therapeutic options for CRPC. Nevertheless, even these approaches cannot durably control tumor growth and there is considerable variability in the nature and duration of responses between different patients (*Tran et al., 2009*; *Scher et al., 2012*; *Davis et al., 2019*). Thus, alternative therapeutic strategies that enhance response to ADT, and thereby prevent or delay PCa progression to CRPC, are essential.

Increasingly, targeting cancer cell metabolism is a focus of research efforts (*Hanahan and Weinberg, 2011*). While fundamental differences in cellular metabolism pathways between normal and malignant cells were detected by Warburg in the 1920s (*Warburg et al., 1927*), clinical targeting of cancer metabolism has not kept pace with the research advances in understanding metabolic features of cancer cells. PCa is mainly dependent on lipid metabolism for energy production (*Liu, 2006*). The overexpression of genes involved in lipid metabolism is characteristic of PCa at both early and advanced stages (*Wu et al., 2014*; *Chen et al., 2018*; *Swinnen et al., 2002*; *Zadra et al., 2013*; *Zadra and Loda, 2018*; *Ettinger et al., 2004*; *Nomura et al., 2011*), while recent proteomic analyses of primary PCa and bone metastases have shown clear associations between levels of lipid metabolic enzymes, PCa initiation and progression (*Iglesias-Gato et al., 2016*; *Iglesias-Gato et al., 2018*). These observations suggest that PCa may be particularly amenable to metabolic targeting strategies. Despite this, the role and complexity of lipid/fatty acid (FA) metabolism in PCa and its potential as a target for therapy remains underexplored, particularly in the context of a more complex tumor microenvironment.

Until recently, most attention has focused on the therapeutic targeting of de novo FA synthesis and, most recently, uptake of FAs in PCa to limit their availability as a source of energy and cell membrane phospholipids (*Zadra et al., 2013*; *Zadra and Loda, 2018*; *Watt et al., 2019*). However, it has become evident in work from our group and others that β-oxidation of FAs, as the ultimate fate of FAs in the energy production cycle, is upregulated in PCa cells, stimulated by a lipid-rich extracellular environment and critical for viability (*Liu, 2006*; *Schlaepfer et al., 2014*; *Balaban et al., 2019*). In this study, we evaluated the targeting of FA β-oxidation (FAO) in patient-derived prostate tumor explants (PDE) to provide the first clinically-relevant evidence that targeting this pathway is efficacious. We subsequently identify DECR1, a rate-limiting enzyme in an auxiliary pathway for polyunsaturated fatty acid (PUFA) β-oxidation, as a promising novel therapeutic vulnerability for PCa. Importantly, we show that DECR1 is an androgen-repressed gene induced in PCa cells in response to ADT and/or AR-targeted therapies, implicating PUFA oxidation as an adaptive survival response that may contribute to emergence of CRPC and treatment resistance.

## Results

### Targeting FA oxidation is efficacious in patient-derived PCa explants

In addition to our recent report of enhanced FAO in PCa cells (*Balaban et al., 2019*), an accumulating body of evidence supports the efficacy of targeting key enzymes involved in FAO using in vitro and in vivo models of PCa (*Itkonen et al., 2017*; *Schlaepfer et al., 2014*; *Flaig et al., 2017*). However, to date there is limited evidence that targeting this pathway would be clinically efficacious, which prompted us to target this pathway in clinical tumors. Using our well-defined patient derived explant (PDE) model that recapitulates the complexity of the clinical tissue microenvironment (*Centenera et al., 2012*), we targeted the rate-limiting enzyme in mitochondrial FAO, carnitine palmitoyltransferase-1 (CPT-1), in cultured PDEs using the chemical inhibitor etomoxir. Consistent with

literature reports, etomoxir had weak activity against the LNCaP PCa cell line in vitro, with an IC50 of 170 µM (*Figure 1—figure supplement 1A*), but was considerably potent in the PDEs, in which a dose of 100 µM inhibited cell proliferation by an average of 48.4 ± 16.6% (n = 13 patients; p<0.05) (*Figure 1A*). Etomoxir effectively inhibited FAO in the tissues, evidenced by a significant decrease in multiple acylcarnitines in the conditioned medium (*Figure 1B*).

In order to prioritize key functional genes involved in PCa progression, we conducted a meta-analysis of the expression of 735 genes involved in lipid metabolism (as identified from REACTOME) in four clinical datasets with malignant and matched normal RNA sequencing data (*Nikitina et al., 2017*; *Ren et al., 2012*; *Ding et al., 2016*). Genes were rank-ordered on the basis of their *meta effect size* scores in PCa malignant versus matched normal tissues (*Figure 1—figure supplement 1B*). The meta-analysis revealed a strikingly consistent deregulation of lipid metabolism genes, including genes involved in FAO (*Figure 1C*), despite the predicted high inter-individual heterogeneity of patient PCa tissues. We conducted disease-relapse survival analysis using TCGA data for each of the top 20 genes from the meta-analysis. This identified *DECR1*, a rate-limiting enzyme in the mitochondrial β-oxidation of polyunsaturated fatty acids, as a robustly overexpressed gene in PCa tissues that is associated with shorter relapse-free survival rates.

## DECR1 is upregulated in clinical prostate tumors

Consistently, *DECR1* mRNA expression was significantly higher in malignant compared to benign prostate tissues in ten independent expression datasets of PCa tissues versus non-malignant tissues (*Figure 1D*, *Figure 1—figure supplement 1C*). Further analysis of the TCGA data revealed increased *DECR1* expression with increased Gleason score or in with advanced disease stage (*Figure 1E*). Consistent with our observation of increased DECR1 mRNA expression in PCa, *DECR1* gene copy gain was evident in several clinical datasets accessed via cBioPortal (*Cerami et al., 2012*; *Figure 1F*). Interestingly, the top three cancer types exhibiting increased *DECR1* copy gain were hormone-dependent tumors (uterine, breast and prostate), suggestive of a relationship between DECR1 expression and hormone signaling (*Figure 1G*). *DECR1* mRNA expression was associated with shorter relapse-free survival rates and overall survival rates (*Figure 1H*), and in the TCGA dataset, *DECR1* amplification was significantly associated with shorter recurrence-free survival rates (*Figure 1I*).

We confirmed overexpression of DECR1 protein in clinical PCa using two independent proteomic datasets (*Figure 2A*). We observed overexpression of DECR1 in PCa tissues (n = 8) compared with benign tissues (n = 3) (*Figure 2B*), and increased expression was evident in high grade versus low grade cancer tissue. Quantitative IHC staining analysis revealed a significant increase of DECR1 expression in malignant *vs* benign tissues. Furthermore, intra-tissue analysis exposed a significant increase of DECR1 expression in malignant regions *vs* benign ones within the same core (*Figure 2C*). DECR1 expression was markedly increased in a panel of hormone-dependent and -independent cancer cell lines compared with non-malignant PNT1 and PNT2 prostate cell lines (*Figure 2D*). Consistent with its function, DECR1 localises to the mitochondria, confirmed using immunocytochemistry and Western blot of nuclear, cytoplasmic and mitochondrial cell fractions (*Figure 2E*). Together, the mRNA and protein findings suggest that expression of DECR1 is closely linked to PCa progression and patient outcomes, and therefore might represent an unexplored therapeutic target.

## DECR1 is a directly androgen-repressed gene in PCa

The relationship between androgen receptor (AR) signaling and lipid metabolic genes is well established. Many studies have reported a marked stimulatory effect of AR on key lipid metabolism pathways either directly or indirectly through activation of a family of transcription factors called sterol regulatory element-binding proteins (SREBPs) (*Butler et al., 2016*). We therefore investigated the relationship between AR and DECR1 using a panel of in vitro, ex vivo and in vivo models. DECR1 expression is notably more abundant in AR-negative cells (PC3) than in AR-expressing cells (*Figure 2D*), consistent with negative regulation of DECR1 expression by AR. We confirmed that androgen (5α-dihydrotestosterone) significantly decreased DECR1 expression in androgen-dependent LNCaP and VCaP cell lines at both mRNA and protein levels (*Figure 3A*). Data mining of publicly available microarray datasets also revealed downregulation of DECR1 in LNCaP cells after treatment with DHT or the synthetic androgen R1881 (*Figure 3—figure supplement 1A*); GSE7868,

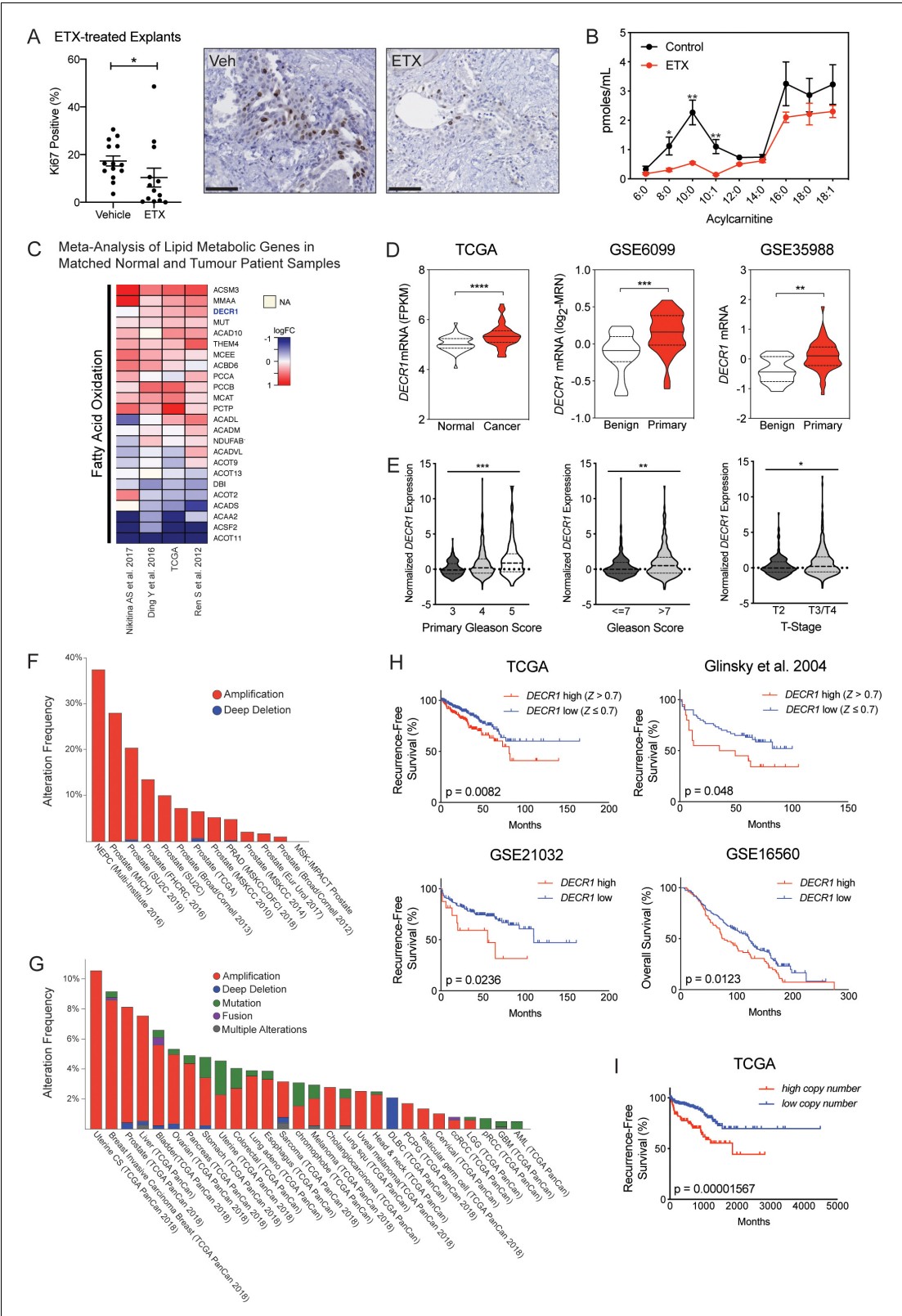

**Figure 1.** Fatty acid β-oxidation genes are overexpressed in prostate cancer and targeting this process is effective in patient-derived human prostatic ex vivo tumor explants. (**A**) Etomoxir reduced cell proliferation in patient-derived human prostatic ex vivo tumor explants. Tissues were treated with 100 μM etomoxir for 72 hr, sections were fixed in formalin, paraffin embedded and stained against the proliferative marker Ki67 (n = 13) (scale bar = 50 μm). (**B**) Etomoxir (100 μM) decreased acylcarnitine species, the products of CPT1 activity. Acylcarnitines secreted to the conditioned medium were

*Figure 1 continued on next page*

*Figure 1 continued*

measured after 72 hr treatment of PDEs (n = 9). (**C**) A meta-analysis of fatty acid oxidation genes using four clinical datasets with malignant and matched normal RNA-sequencing data (n = 122). Genes were rank-ordered on the basis of their *meta effect size* scores in PCa malignant tissues versus matched normal tissues. (**D**) Violin plots demonstrate DECR1 mRNA overexpression in PCa primary/malignant tissues compared to normal/benign tissues in three independent datasets. (**E**) DECR1 mRNA expression is associated with PCa primary Gleason score, total Gleason score (>7) and diseases stage (T-stage). Data were extracted from TCGA PCa dataset. (**F**) Histogram displaying DECR1 mutation and copy-number amplification frequency across 13 PCa genomic datasets, and (**G**) across 28 tumor types. Histograms were obtained from CbioPortal platform. (**H**) DECR1 mRNA expression is associated with shorter relapse-free survival in TCGA PCa, *Glinsky et al., 2005* and GSE21032 datasets, and shorter overall survival rates in GSE16560 dataset. (**I**) DECR1 copy number amplification frequency is associated with shorter relapse-free survival in TCGA PCa dataset. Data in (**A**) are represented as as the mean ± s.e.m and were statistically analysed using a Wilcoxon matched-pairs signed rank test. Data in (**B**) are represented as the mean ± s.e.m and were statistically analysed using two-tailed Student's *t*-test. Data in (**D**) and (**E**) are represented as violin plots in GraphPad prism: the horizontal line within the violin represents the median, and were statistically analysed using a Mann-Whitney two-tailed *t*-test. Data in (**H**) and (**I**) were statistically analysed using a two-sided log-rank test. *p<0.05, **p<0.01, ***p<0.001 and ****p<0.0001.

The online version of this article includes the following figure supplement(s) for figure 1:

**Figure supplement 1.** Fatty acid metabolism is consistently altered in clinical prostate tumors.

GSE22606). In contrast to the effect of androgens, AR-targeted therapies increase DECR1 expression. LNCaP and VCaP cell treatment with the androgen antagonist enzalutamide (ENZ) significantly increased DECR1 expression at both mRNA and protein levels (*Figure 3B*). In vivo, LNCaP tumors exhibited increased DECR1 expression in mice treated with ENZ (10 mg/kg) or castration, which was enhanced further in mice treated with both ENZ and castration (*Figure 3C*). Our observations were supported by published microarray datasets which showed that treatment of LNCaP or VCaP cells with ENZ increases *DECR1* mRNA expression (*Figure 3—figure supplement 1B*; GSE69249). In vivo, castration of mice increased DECR1 expression in prostate (*Figure 3—figure supplement 1C*; GSE5901), while in LNCaP/AR xenografts treatment with the AR antagonist ARN-509 (apalutamide) for 4 days significantly increased DECR1 expression (*Figure 3—figure supplement 1D*; GSE52169). To confirm androgenic regulation of DECR1 in a clinical context, we validated these data using PDEs. ENZ treatment of PDEs significantly increased DECR1 expression whereas, as expected, mRNA levels of the well-characterized AR target genes *KLK3* and *KLK2* were decreased (*Figure 3D, E*). To determine whether AR directly represses *DECR1*, we interrogated published chromatin immunoprecipitation (ChIP) sequencing data. In VCaP cells, AR bound strongly to the *DECR1* promoter in response to DHT treatment, but not when co-treated with AR antagonists (*Figure 3F*; GSE55064). Moreover, AR binding was enriched at the *DECR1* promoter in benign and malignant prostate tissues (*Figure 3—figure supplement 1E*; GSE56288). A site-specific ChIP-qPCR assay revealed DHT-stimulated AR occupancy at this region in LNCaP cells (*Figure 3G*). Collectively, these data reveal *DECR1* as a novel AR-repressed gene.

## Targeting DECR1 disrupts PUFA oxidation in PCa cells

FAs are metabolized mainly in mitochondria through the β-oxidation process to generate acetyl-CoA, which enters the tricarboxylic acid cycle (TCA) and produces ATP and NADH as energy for the cell. Unlike saturated FAs, all unsaturated FAs with double bonds originating at even-numbered positions, and some unsaturated FAs with double bonds originating at odd-numbered positions, require three auxiliary enzymes to generate intermediates that are harmonious with the standard β-oxidation pathway (*Hiltunen and Qin, 2000*; *Shoukry and Schulz, 1998*): Enoyl CoA isomerase (ECI1), 2,4 Dienoyl-CoA reductase (DECR1) and Dienoyl CoA isomerase (ECH1) (*Figure 4A*). DECR1 catalyses the rate limiting step in this pathway (*Alphey et al., 2005*). Given the critical role of DECR1 in PUFA metabolism, we studied the consequences of DECR1 downregulation on β-oxidation of PUFAs in PCa cells. DECR1 knockdown was achieved successfully (>80% downregulation) using two different siRNAs (*Figure 4B*). DECR1 knockdown resulted in an increase in linoleic acid (*Figure 4C*) as well as an accumulation of 2-trans,4-cis-decadienoylcarnitine (acylcarnitine 10:2; *Figure 4D*), an intermediate of linoleic acid metabolism, indicating incomplete PUFA β-oxidation. Mitochondrial β-oxidation provides reducing equivalents that drive ATP production. LNCaP cells increased ATP levels in response to exogenous linoleic acid supplementation when cultured in glucose-free media containing the lipase inhibitor diethylumbelliferyl phosphate (DEUP), to prevent the cells from using intracellular FAs (*Figure 4E*). However, cells transfected with DECR1-targeting

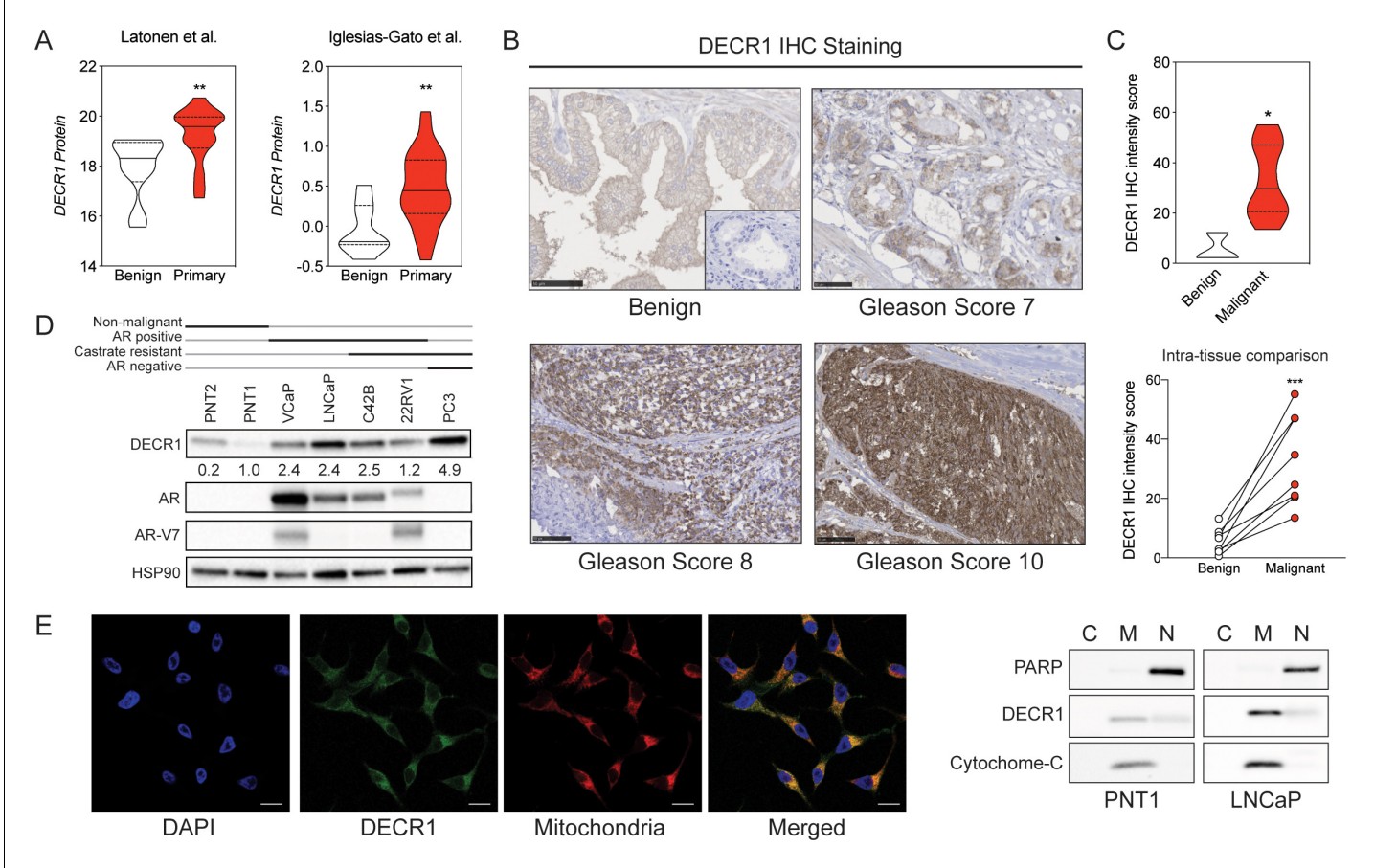

**Figure 2.** DECR1 protein in overexpressed in malignant prostate cells/tissues. (**A**) Violin plots of DECR1 protein overexpression in primary PCa tissues compared to benign prostate tissues in two independent datasets. (**B**) Representative DECR1 IHC staining of benign prostate tissues and PCa tissues (negative control stain included in bottom right box). Scale bar, 50 μm. (**C**) Violin plot of DECR1 protein expression in a validation cohort consisting of benign prostate tissues (n = 3) and PCa tissues (n = 8) (*top panel*). Intra-tissue IHC analysis of DECR1 expression in PCa tissues (n = 8) (*bottom panel*). (**D**) DECR1 protein expression in non-malignant prostate cell lines (PNT1 and PNT2) and PCa cell lines (LNCaP, VCaP, 22RV1, C42B and PC3). (**E**) Immunocytochemistry staining of LNCaP cells to determine the subcellular localization of DECR1: nuclei were labelled using DAPI; mitochondria were labelled using MitoTracker Red; and DECR1 proteins were labelled using Alexa Fluor 488 secondary antibody, (Scale bar = 10 μm). Immunoblot of PNT1 and LNCaP cells separated into cytosolic, mitochondrial and nuclear fractions and incubated with poly (ADP-ribose) polymerase (PARP) and cytochrome-C antibodies to mark nuclear and mitochondrial fractions. Data are represented as violin plots in GraphPad prism: the horizontal line within the violin represents the median. Statistical analysis was performed using a Mann-Whitney two-tailed t-test (A and C *top panel*), or two-tailed paired t-test (C *bottom panel*): *$p<0.05$, **$p<0.01$ and ***$p<0.001$.

siRNAs failed to increase ATP levels with linoleic acid supplementation (*Figure 4E*), indicating impaired capacity to metabolize PUFAs.

Next, we employed extracellular flux analysis to determine the intrinsic rate and capacity of PCa cells to oxidise PUFAs in conditions where other exogenous substrates were limiting. Exogenous linoleic acid stimulated basal oxygen consumption rates (OCR), as a measure of mitochondrial oxidative phosphorylation, and maximal respiration, ATP production, and mitochondrial spare capacity (*Figure 4F*, *Figure 4—figure supplement 1A*) as determined by consecutive cell exposure to respiration chain inhibitors and uncouplers. This supports the observed increased in total ATP levels in response to linoleic acid supplementation (*Figure 4E*). Importantly, DECR1 knockdown prevented the exogenous linoleic acid induction of basal and maximal respiration, ATP production, and mitochondrial spare capacity (*Figure 4F*, *Figure 4—figure supplement 1A*). In contrast, DECR1 knockdown has no impact on mitochondrial metabolism of the saturated FA, palmitate (*Figure 4G*). Further, DECR1 knockdown increased glycolysis, as determined by ECAR (*Figure 4H*, *Figure 4—figure supplement 1B*), as well as decreased glucose and fructose concentrations (*Figure 4I*), to sustain

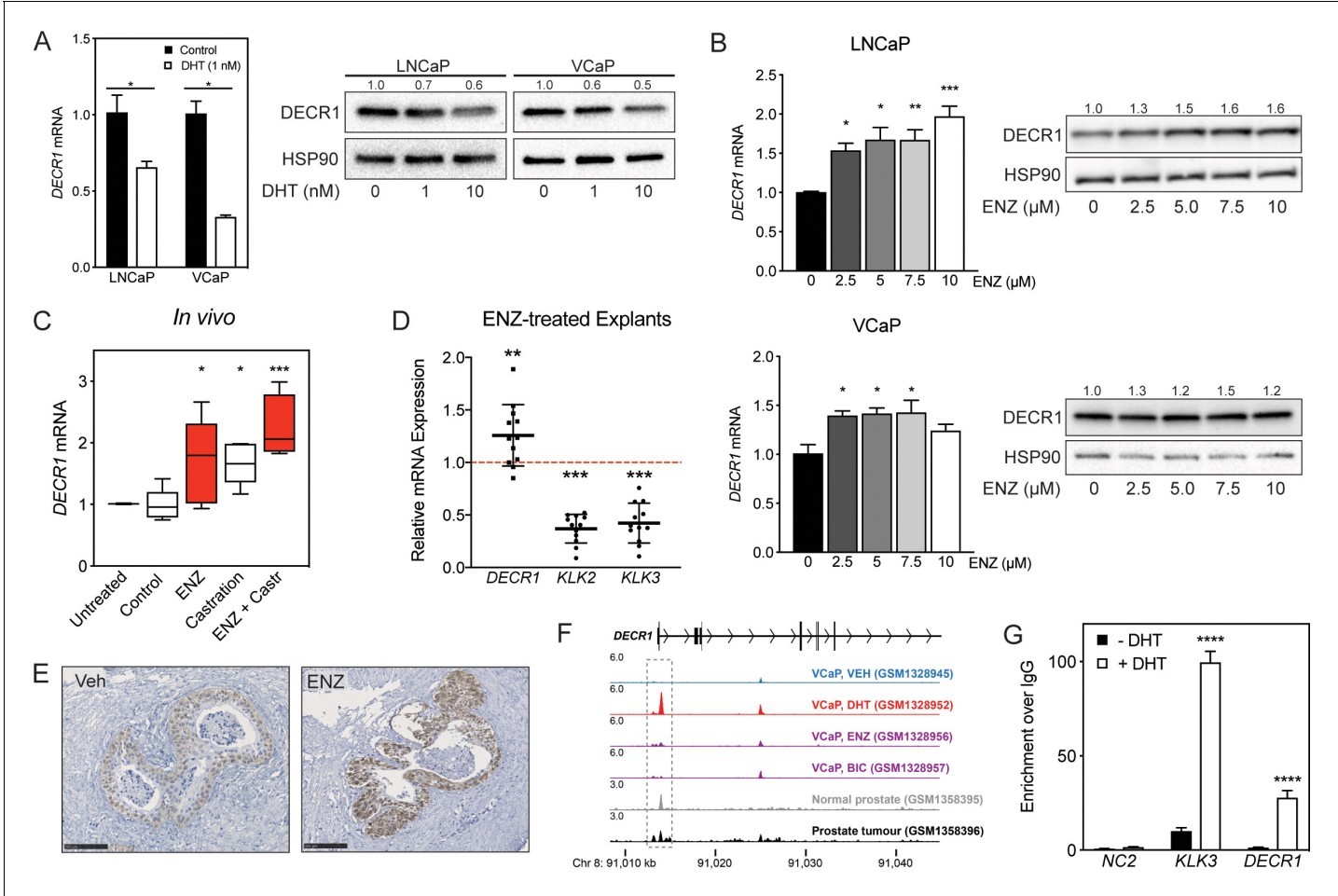

**Figure 3.** DECR1 is an androgen-repressed gene. (**A**) DECR1 mRNA and protein was measured by qRT-PCR and western blot analysis after PCa cell treatment with dihydrotestosterone (DHT), or (**B**) enzalutamide (ENZ). Relative mRNA expression of *DECR1* was calculated using comparative CT method, where the cells treated with vehicle (control) were set to one and normalised to the geometric mean CT value of *GUSB* and *L19* (housekeeping genes). Densitometry quantification of relative DECR1 protein expression was normalized to the HSP90 internal control. *n* = 3 independent experiments. (**C**) qRT-PCR analysis of *DECR1* mRNA expression in LNCaP-derived tumors treated with enzalutamide (ENZ) and/or castration (Castr). (**D**) qRT-PCR analysis of *DECR1* and androgen regulated genes *KLK2* and *KLK3* mRNA expression in a cohort of 10 patient-derived human prostatic ex vivo tumor explants treated with enzalutamide (ENZ, 10 µM). Relative mRNA expression was calculated using comparative CT method, where the matched untreated tissue from the same patient was set to one and normalized to the geometric mean CT value of *TUBA1B, PPIA* and *GAPDH*. (**E**) Representative DECR1 IHC staining of patient-derived human prostatic ex vivo tumor explants treated with enzalutamide (ENZ, 10 µM). Scale bar, 100 µm. (**F**) AR ChIP-sequencing data from VCaP cells (top panel), normal human prostate and primary human prostate tumor specimens (bottom panel). Data from GSE55064 and GSE56288. (**G**) ChIP-qPCR analysis demonstrates AR binding at *DECR1* locus in LNCaP cells after treatment with DHT. Data in bar graphs (A, B and G) are represented as the mean ± s.e.m. Data in (**C**) are represented as box plots using the Tukey method in GraphPad prism. Statistical analysis was performed using two-tailed Student's *t*-test (A, C, D and G) or one-way ANOVA, followed by Dunnett's multiple comparisons test (B): *p<0.05, **p<0.01, ***p<0.001 and ****p<0.0001.

The online version of this article includes the following figure supplement(s) for figure 3:

**Figure supplement 1.** Androgenic regulation of DECR1 expression.

TCA cycle intermediate levels (*Figure 4J*) as a consequence of disruption of PUFA β-oxidation. While DECR1 exhibited selectivity in inhibition of PUFA metabolism, as expected, etomoxir inhibited both PUFA and saturated fatty acids (*Figure 4—figure supplement 1C*). There was no effect of ETX treatment on DECR1 expression (*Figure 4—figure supplement 1D*). Collectively, these results demonstrate that DECR1 is critical for PUFA metabolism in LNCaP PCa cells.

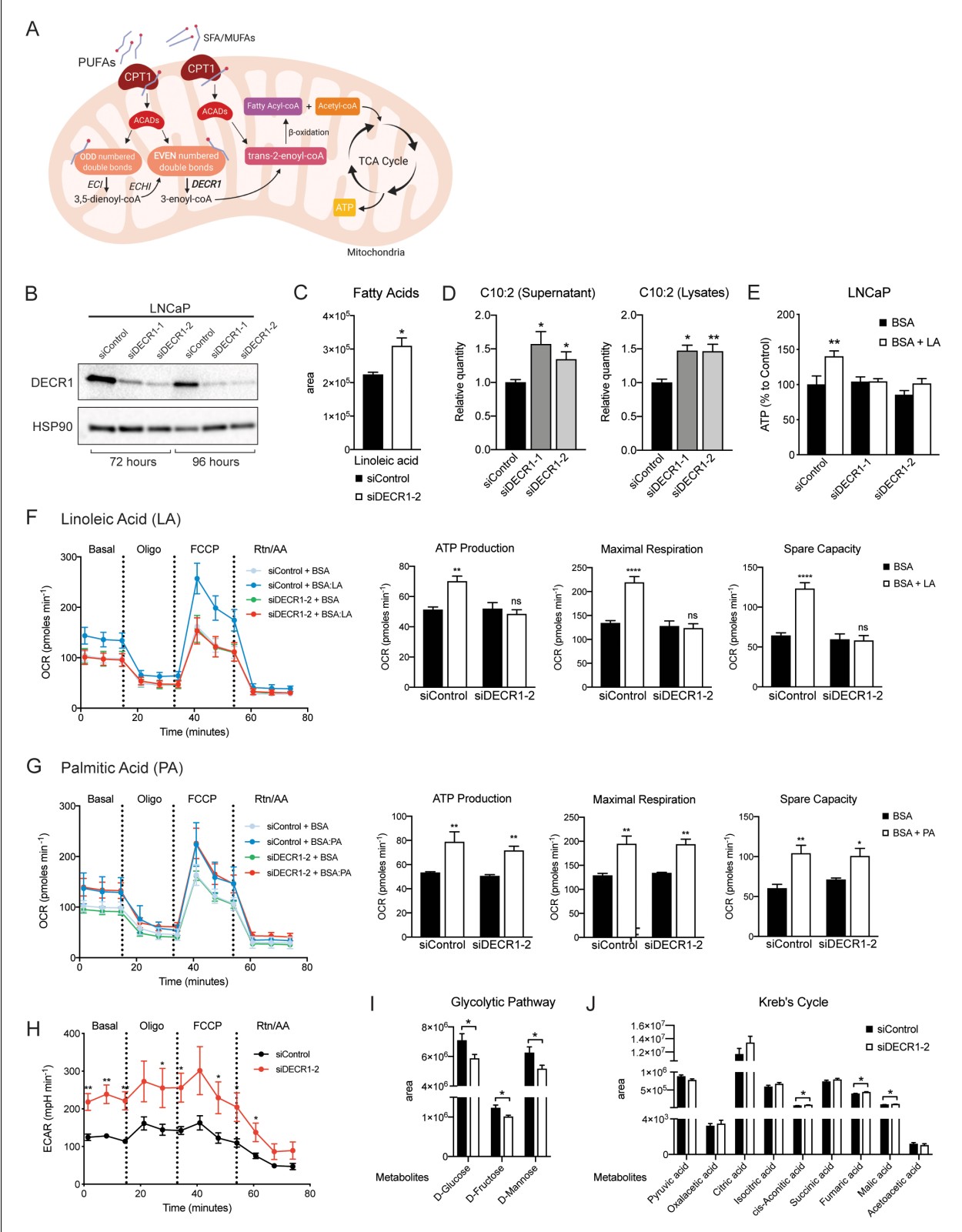

**Figure 4.** DECR1 knockdown interrupts PUFA β-oxidation in PCa cells. (**A**) Schematic of DECR1 function in fatty acid (FA) β-oxidation. In order to translocate FAs into the mitochondria, CPT1 converts long-chain acyl-CoA species to their corresponding long-chain acylcarnitine species. This is followed by a dehydrogenation step mediated by acyl CoA dehydrogenase (ACAD) to generate trans-2-enoyl-CoA, the only intermediate that can be processed by downstream enzymes in the β-oxidation process. Many FAs have unsaturated bonds either on an odd-numbered carbon or in the cis-

*Figure 4 continued on next page*

*Figure 4 continued*

configuration, resulting in the generation of enoyl-CoA intermediates that cannot be directly processed via the downstream β-oxidation enzymes. These FAs require the activity of 3 auxiliary enzymes, ECI1, ECH1 and DECR1 in order to form trans-2-enoyl-CoA before undergoing β-oxidation. DECR1 catalyzes the conversion of either 2-trans,4-cis-dienoyl or 2-trans,4-trans-dienoyl-CoA to 3-trans-enoyl-CoA. A complete cycle of β-oxidation results in the release of the first two carbon units as acetyl-CoA, and a fatty-acyl-CoA minus two carbons. The acetyl-CoA enters the TCA cycle to produce energy (ATP). The shortened fatty-acyl-CoA is processed again starting with the ACADs to form trans-2-enoyl-CoA either directly or with the aid of the auxiliary enzymes depending on the presence of double bonds. This process continues until all carbons in the fatty acid chain are turned into acetyl-CoA. (B) DECR1 protein expression after 72 hr or 96 hr siRNA transfection. Densitometry quantification of relative DECR1 protein expression was normalized to the HSP90 internal control. (C) Linoleic acid level in LNCaP cells quantified in following 96 hr DECR1 knockdown using GC QQQ targeted metabolomics. (D) Relative quantities of the C10:2 acylcarnitine species in LNCaP cell conditioned medium (left) or cell lysates (right) (n = 3). (E) Quantification of ATP levels in LNCaP cell lysates. LNCaP cells were transfected with DECR1 siRNAs for 48 hr and then starved in no-glucose medium and treated with the lipolysis inhibitor DEUP (100 µM) in the presence (BSA-LA) or absence (BSA) of the PUFA linoleic acid for 48 hr before measuring ATP levels. (F) Oxygen consumption rate (OCR) was assessed in LNCaP cells supplemented with the PUFA linoleic acid (LA) or (G) the saturated fatty acid palmitic acid (PA). Each data point represents an OCR measurement. ATP production, maximal mitochondrial respiration and mitochondrial spare capacity were assessed. (H) Extracellular acidification rate (ECAR) was assessed in LNCaP cells. Each data point represents an ECAR measurement. For experiments (F-H) LNCaP cells were transfected with DECR1 siRNAs for 72 hr, then starved in substrate limited medium for 24 hr; the assay was run in FAO assay medium. (I and J) Metabolites were quantified in LNCaP cells following 96 hr DECR1 knockdown using GC QQQ targeted metabolomics. Data in bar graphs are represented as the mean ± s.e.m (n = 3). Statistical analysis was performed using two-tailed Student's *t*-test: *p<0.05, **p<0.01 and ****p<0.0001.

The online version of this article includes the following figure supplement(s) for figure 4:

**Figure supplement 1.** Effects of DECR1 on prostate cancer cellular metabolism.

## Targeting DECR1 suppresses PCa oncogenesis

The consistently increased expression of DECR1 in PCa tissue and its association with shorter-relapse times and survival rates (*Figure 1* and *2*), taken together with its impact on PUFA metabolism (*Figure 4*), suggested that it may contribute to PCa cell viability and invasive behaviour. We evaluated the impact of DECR1 downregulation or overexpression on various oncogenic properties of PCa cells using a series of in vitro and in vivo experiments. While there was no effect of DECR1 downregulation on the non-malignant prostate cell line PNT1, a significant attenuation of PCa proliferation and induction of cell death was observed in a panel of PCa lines (*Figure 5A*), comprising androgen-dependent (VCaP and LNCaP), CRPC (22RV1 and V16D) and acquired ENZ-resistant cells (MR49F). Notably, this effect on PCa cell viability was lost when cells were cultured in lipid-depleted media (*Figure 5—figure supplement 1*), suggesting that the observed effect is due to interference with FA metabolism. Likewise, stable DECR1 knockdown using a short hairpin vector attenuated LNCaP cell line viability and induced cell death (*Figure 5B*) In contrast, stable DECR1 overexpression significantly enhanced LNCaP cell viability (*Figure 5C*) and colony formation ability (*Figure 5D*), while stable DECR1 knockdown markedly decreased colony formation (*Figure 5E*). DECR1 knockdown also decreased LNCaP growth in 3D spheroids (*Figure 5F*), which better mimic in vivo conditions than 2-dimensional cell culture (*Duval et al., 2017*). In addition, DECR1 knockdown reduced LNCaP, 22RV1 and MR49F cell migration by ~50% (*Figure 5G*) and 22RV1 invasion by ~65% (*Figure 5H*). In vivo, LNCaP cells stably depleted of DECR1 showed highly variable growth rates in a subcutaneous model (*Figure 5—figure supplement 2A*), but inspection of the resultant tumors revealed significantly reduced cellular proliferation compared to control cells, concomitant with reduced DECR1 expression (*Figure 5I*, *Figure 5—figure supplement 2B*). To study the effect of DECR1 downregulation on PCa in the prostate microenvironment, we undertook a second study using LNCaP orthotopic xenografts. DECR1 knockdown significantly retarded tumor growth (*Figure 5K*, (*Figure 5—figure supplement 2C,D,E*), and significantly inhibited lung metastasis in the orthotopic tumor model (*Figure 5L*).

## DECR1 targeting induces lipid peroxidation and cellular ferroptosis

DECR1 knockdown resulted in inhibition of PUFA β-oxidation and led to accumulation of PUFAs in phospholipids (*Figure 6A*). Inspection of the phospholipid profile revealed accumulation of PUFAs in the PC, PI and PS classes (*Figure 6B*) with no impact on total saturated or MUFA phospholipids (*Figure 6—figure supplement 1A*). PUFA are highly susceptible to peroxidation, so we next assessed the effect of DECR1 knockdown on lipid peroxidation. DECR1 knockdown increased levels of

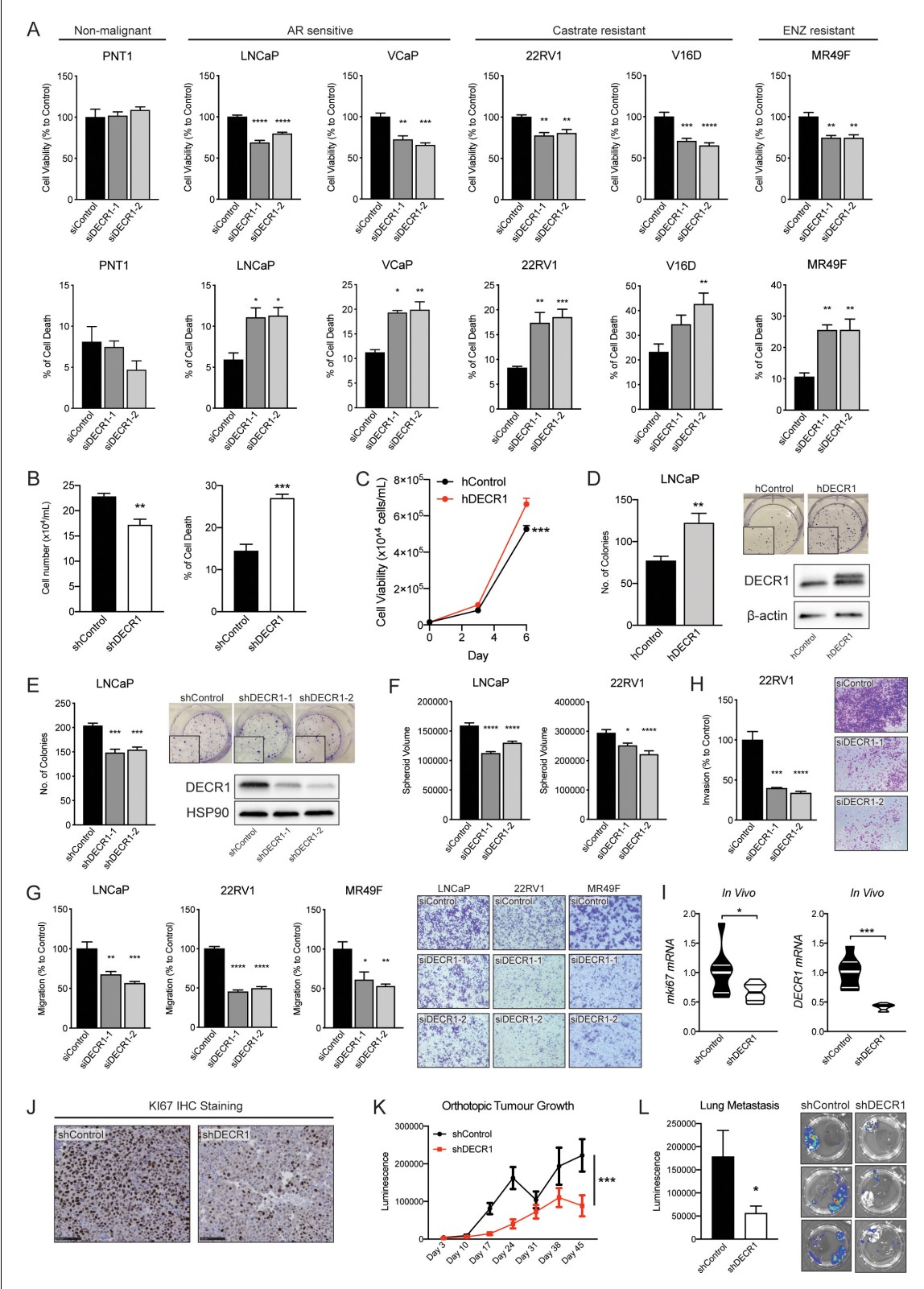

**Figure 5.** DECR1 knockdown suppresses oncogenic phenotypes of PCa cells. (**A**) Cell viability after DECR1 knockdown in non-malignant PNT1 prostate cells; hormone-responsive PCa cell lines (LNCaP and VCaP); castrate-resistant V16D and 22RV1 cell lines and enzalutamide-resistant MR94F cells cultured in full serum media. (**B**) Cell viability and cell death of stable DECR1 knockdown LNCaP cells cultured in full serum media. (**C**) Cell viability of stable DECR1-overexpressed LNCaP cells cultured in full serum media. Cell viability and cell death were measured using trypan blue exclusion

*Figure 5 continued on next page*

*Figure 5 continued*

following 96 hr DECR1 knockdown. Percentages are represented relative to the control siRNA; *n* = 3 independent experiments per cell line. (D) Clonogenic cell survival of LNCaP cells was assessed using colony formation assay. Stable DECR1-overexpressed cells or (E) stable DECR1 knockdown was achieved using two different short hairpin (sh) vectors and DECR1 expression was confirmed using western blot. Cells were cultured for 2 weeks, washed with PBS, fixed with paraformaldehyde and stained with 1% crystal violet for 30 min. Colonies with more than 50 cells were counted manually; data shown is representative of *n* = 2 independent experiments. (F) LNCaP and 22RV1 cell growth in 3D spheres. Spheroids were prepared using the hang drop assay following 48 hr DECR1 knockdown. Spheroid volumes were determined after five days of culturing the cells in 20 µl drops; at least 25 spheres per cell line were assessed using the ReViSP software, *n* = 3 independent experiments per cell line. (G) LNCaP, 22RV1 and MR49F cell migration and (H) 22RV1 cell invasion were assessed using a transwell migration/invasion assay. Cells were transfected with DECR1 siRNA or control siRNA for 48 hr prior to the assay; data shown is representative of *n* = 3 independent experiments. (I) Violin plots of mKi67 and DECR1 mRNA expression in subcutaneous LNCaP tumors (n = 5 mice, shControl; n = 4 mice, shDECR1). (J) Representative Ki67 IHC staining of subcutaneous LNCaP tumors. Scale bar, 100 µm. Data in bar graphs are represented as the mean ± s.e.m. Statistical analysis was performed using one-way ANOVA, followed by Dunnett's multiple comparisons test: *p<0.05, **p<0.01, ***p<0.001 and ****p<0.0001. (K) Tumor growth of intraprostatically injected LNCaP cells (shControl and shDECR1). (J) Lung luminescence readings following DECR1 knockdown in mice. Data are presented as mean ± s.e.m. Statistical analysis was performed using two-way ANOVA or two-tailed student's t-test: *p<0.05 and ***p<0.001.

The online version of this article includes the following figure supplement(s) for figure 5:

**Figure supplement 1.** Depletion of extracellular lipids prevents antiproliferative effects of DECR1 in prostate cancer cells.
**Figure supplement 2.** DECR1 suppresses growth of prostate tumor xenografts in mice.
**Figure supplement 3.** The sequence of the DECR1 shRNA and hDECR1 vectors.

malondialdehyde, a marker of lipid peroxidation (*Gaweł et al., 2004*; *Figure 6C*). In contrast, DECR1 overexpression markedly decreased cellular malondialdehyde levels (*Figure 6C*). We observed enhanced mitochondrial oxidative stress measured using MitoSOX, a mitochondrial super-oxide indicator (*Figure 6D*), in response to DECR1 knockdown. Moreover, DECR1 knockdown resulted in significant accumulation of phospholipid hydroperoxides, a hallmark of ferroptosis (*Figure 6E*). In contrast, DECR1 overexpression significantly decreased mitochondrial oxidative stress under basal (*Figure 6F*) and linoleic acid-induced conditions (*Figure 6G*). Lipid peroxidation is a major driver of ferroptosis, an iron-dependent non-apoptotic form of cell death (*Dixon et al., 2012*). Cell treatment with the ferroptosis inhibitors, deferoxamine or ferrostatin, abolished the effect of DECR1 knockdown on PCa cell death (*Figure 6H–I*), while cell treatment with the ferropto-sis inducers, erastin, FIN56 or ML210, enhanced the cytotoxic action of DECR1 downregulation (*Figure 6J–K*, *Figure 6—figure supplement 1B*). Supporting ferroptosis as the cell death mecha-nism, DECR1 knockdown did not induce apoptosis (*Figure 6—figure supplement 2A*), and the apo-ptosis inhibitor ZVAD did not rescue the cells from cell death induced by DECR1 depletion (*Figure 6—figure supplement 2B*). Collectively, these results suggest that DECR1 expression pro-tect cells from oxidative stress and that DECR1 knockdown-induced cell death is primarily mediated by induction of ferroptosis.

## Discussion

Metabolic rewiring is both a hallmark feature of cancer cells and a promising therapeutic vulnerabil-ity. Several anabolic and catabolic metabolism pathways have been explored, however, few agents have been investigated clinically. This can at least partly be explained by the relatively recent appre-ciation of cancer metabolism as a target, the high toxicity, particularly hepatotoxicity, expected to be associated with targeting certain metabolic pathways, the predictable metabolic heterogeneity within and between patients and the lack of intermediate pre-clinical models that can predict clini-cally efficacious outcomes. Previous research has largely focused on studying and targeting FA syn-thetic pathways in PCa. Major lipogenic enzymes such as ATP citrate lyase (ACLY), acetyl-CoA carboxylase (ACC) and fatty acid synthase (FASN) are all overexpressed in PCa compared to benign tissue (*Wu et al., 2014*; *Rossi et al., 2003*; *Shurbaji et al., 1996*). While many first-generation FA synthesis inhibitors (e.g. FASN inhibitors) showed promising preclinical efficacy against PCa, unfavor-able drug solubility and pharmacokinetics profiles, off-target effects and side effects including weight loss have hindered clinical development of this agent class (*Zadra et al., 2013*). In addition to de novo synthesis of FAs, PCa cells depend on lipid uptake from the circulation, and from stromal adipocytes (*Watt et al., 2019*; *Kuemmerle et al., 2011*; *Gazi et al., 2007*). We showed previously that extracellular FAs are the major contributor to lipid synthesis in PCa (*Balaban et al., 2019*).

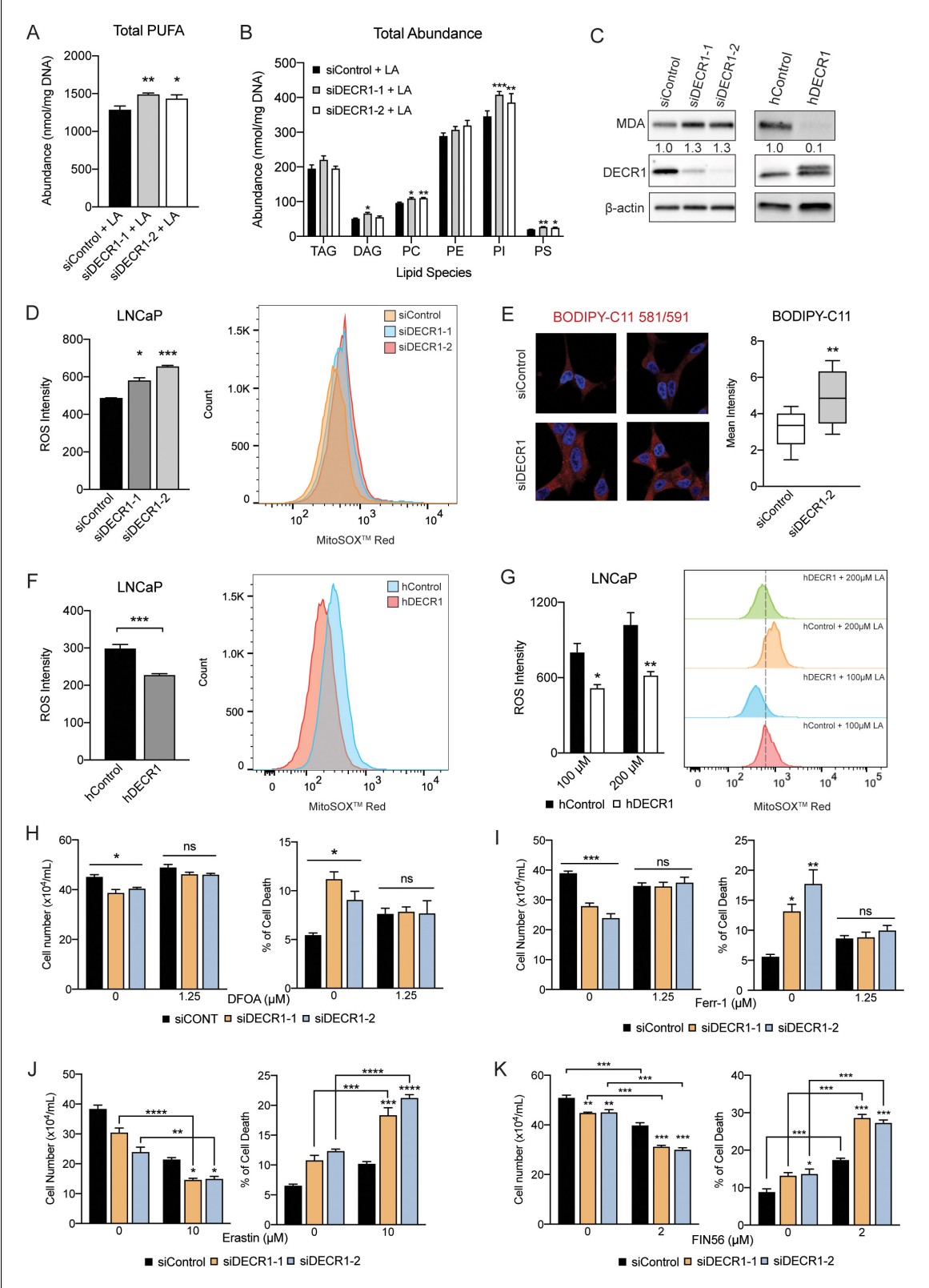

**Figure 6.** DECR1 knockdown induces PUFA accumulation, lipid peroxidation and ferroptosis. (**A**) Abundance of total PUFAs and (**B**) total abundance of lipid species in phospholipids from control and DECR1 knockdown cells supplemented with linoleic acid (LA). (**C**) Malondialdehyde (MDA), an oxidative stress marker, was measured by western blot in LNCaP cells transfected with DECR1 siRNAs and in DECR1-overexpressed LNCaP cells. (**D**) Mitochondrial superoxide levels were quantified following 96 hr DECR1 knockdown using MitoSOX red stain. Fluorescent intensity was quantified using

*Figure 6 continued on next page*

*Figure 6 continued*

flow cytometry and ROS levels presented as mean fluorescent intensity. (E) Fluorescent images of BODIPY-C11 stained LNCaP cells following DECR1 knockdown (left). Fluorescent intensity was quantified using ImageJ and presented as mean fluorescent intensity (right). (F) Mitochondrial superoxide levels were quantified in stable DECR1-overexpressing LNCaP cells using MitoSOX red stain under basal or (G) linoleic acid (100 µM or 200 µM) conditions. Fluorescence intensity was quantified using flow cytometry and ROS levels were presented as mean fluorescent intensity. Cell viability of LNCaP cells after 48 hr DECR1 knockdown, treated with (H) deferoxamine (DFOA, 1.25 µM), (I) ferrostatin (Ferr-1, 1.25 µM), (J) erastin (10 µM), or (K) FIN56 (2 µM). Data in bar graphs are represented as the mean ± s.e.m. Statistical analysis was performed using two-tailed Student's *t*-test (A-G) or one-way ANOVA, followed by Holm-Sidak's multiple comparisons test (H-K): ns, not significant, *p<0.05, **p<0.01, ***p<0.001 and ****p<0.0001. TAG: triacylglycerol; DAG: diacylglycerol; PC: phosphotidylcholine; PE: phosphotidylethanolamine; PI: phosphatidylinositol; PS: phosphotidylserine.

The online version of this article includes the following figure supplement(s) for figure 6:

**Figure supplement 1.** Depletion of DECR1 sensitizes prostate cancer cells to ferroptosis inducing agents.
**Figure supplement 2.** Targeting DECR1 does not induce apoptosis of prostate cancer cells.

Moreover, targeting FA uptake using an antibody against CD36, a major transporter for exogenous FAs into the cells, reduced cancer severity in patient-derived xenografts, and *CD36* deletion slowed cancer progression in prostate-specific *PTEN*$^{-/-}$ mice. However, it is increasingly evident that PCa exhibits plasticity in attaining FAs and that crosstalk between de novo synthesis and FA uptake requires dual targeting of the two pathways to achieve maximal efficacy (*Watt et al., 2019*), an approach that would likely be associated with greater toxicity. In this study, we focused on another understudied aspect of FA metabolism in PCa, β-oxidation, to evaluate its therapeutic potential.

We showed previously (*Balaban et al., 2019*), as have others (*Schlaepfer et al., 2014*; *Schlaepfer et al., 2015*), that PCa cells exhibit increased FAO compared to prostate epithelial PNT-1 cells, or benign epithelial cells BPH-1 and WPMY-1. This metabolic phenotype is a vulnerability for PCa cells (*Schlaepfer et al., 2014*; *Itkonen et al., 2017*; *Flaig et al., 2017*). FAO inhibitors such as etomoxir, perhexiline or ranolazin inhibited tumor growth in vitro and in vivo (*Schlaepfer et al., 2014*; *Itkonen et al., 2017*; *Flaig et al., 2017*) and sensitized cells to ENZ treatment (*Itkonen et al., 2017*; *Flaig et al., 2017*). However, all of these studies were undertaken using immortalized cell line models of PCa, and as cancer metabolism is markedly influenced by the tumor microenvironment (*Martinez-Outschoorn et al., 2012*; *Nieman et al., 2011*; *Nassar et al., 2018*), employing preclinical models and primary tissues that retain the complexity of this microenvironment as a stepping stone to clinical trials may accelerate clinical translation and avoid futile targeting strategies or agents. In this study, we evaluated the efficacy of the FAO inhibitor, etomoxir, using our established PDE model. Remarkably, etomoxir inhibited effectively cell proliferation in PDEs (*Figure 1A*), strengthening the case that targeting FA oxidation may be a promising clinical strategy. Interestingly, etomoxir was more potent in inhibition of cell proliferation in PDEs than in vitro 2-dimensional growth of LNCaP cells, emphasizing that in vitro models based on 2D cultured cells alone are suboptimal when evaluating anti-metabolism agents. This problem is compounded by the fact that standard growth media is rich in sugar and proteins, but contains low levels of lipids. As the clinical development of etomoxir was terminated due to severe hepatotoxicity associated with treatment (*Holubarsch et al., 2007*), we sought to identify new β-oxidation targets in PCa. As DECR1 is a directly androgen-repressed gene, its expression increases after castration or treatment with anti-androgens and is hypothesized to maintain tumor cell survival under castration conditions. Androgen-repressed genes are markedly understudied compared with androgen-induced genes, despite the fact that they possibly mediate adaptive survival pathways when androgen signalling is perturbed, and have already yielded novel therapeutic targets (*Kregel et al., 2013*; *Tse et al., 2017*).

Surprisingly, very little is known about the biological role of DECR1 in cancer. Human DECR1 deficiency is lethal, with patients exhibiting hypocarnitinemia, decreased cellular oxygen consumption, increased lactic acidosis, and unusual accumulation of FA intermediates in urine and blood due to incomplete β-oxidation (*Roe et al., 1990*; *Houten et al., 2014*). DECR1-null mice exhibit impaired lipid metabolism, hypoglycemia and activation of ketogenesis, and cold intolerance (*Miinalainen et al., 2009*; *Mäkelä et al., 2019*). These phenotypes highlight the critical role of DECR1 in lipid metabolism. We confirmed the metabolic activities of DECR1 in PCa cells using a panel of metabolomic and lipidomic analyses. DECR1 knockdown increased levels of certain acylcarnitine species, indicating inhibition of β-oxidation. These results are consistent with previous studies reporting acylcarnitine accumulation in *Decr1*$^{-/-}$ mice and *DECR1*-deficient patients (*Roe et al.,*

1990; *Miinalainen et al., 2009*). We showed that DECR1 knockdown selectively inhibited PUFA catabolism, accompanied by an increase in glycolysis rate, which is also consistent with previous reports of FAO inhibition or impaired mitochondrial function leading to enhanced glucose uptake and glycolysis (*Schlaepfer et al., 2015*; *Houten et al., 2014*). Lipidomic analysis showed that DECR1 knockdown increased the abundance of PUFAs and certain lipids, particularly PE and PI phospholipid species, accompanied by increased levels of mitochondrial oxidative stress and particularly lipid peroxidation. In contrast to MUFAs, PUFAs are highly susceptible to peroxidation, thereby enhancing free radical generation and accumulation of toxic lipid peroxides (*Das, 2011*; *Magtanong et al., 2019*). Consistent with a role for PUFA oxidation in DECR1 function, ectopic DECR1 overexpression decreased mitochondrial oxidative stress. An important cellular protective response to excess intracellular lipid peroxides is the induction of ferroptosis, an iron-dependent form of cell death that is triggered by lipid peroxidation. Here, we show that the ferroptosis inhibitors ferrostatin and deferoxamine or the ferroptosis inducers erastin, FIN56 and ML210 abolished and augmented the effect of DECR1 on PCa cell death, respectively. These findings are consistent with a mechanism by which DECR1 knockdown-induced cell death is a ferroptosis-mediated process caused by PUFA accumulation, a conclusion that was independently validated in human prostate cancer during revision of this article (*Blomme et al., 2020*). It is therefore possible that PCa cells commonly select for DECR1 overexpression, not only to enhance ATP production to fulfil energy requirements, but also to protect cells from the tumoricidal effects of excess PUFAs.

PUFAs are essential FAs that cannot be synthesized in mammals and are only obtained from the diet. Dietary fat not only promotes obesity, but also PCa progression and disease aggressiveness. Several preclinical PCa studies have compared high-fat (or Western-style diets) versus low fat diet and reported that the former promotes AKT and ERK activity, tumor growth, tumor incidence in genetically engineered/transgenic mouse models, tumor progression to CRPC and metastasis (reviewed in *Narita et al., 2019*). Clinical case-control studies indicated saturated fat is associated with an increased risk of advanced PCa (*Bairati et al., 1998*; *Stéfani et al., 2000*; *Slattery et al., 1990*; *Whittemore et al., 1995*). Several underlying mechanisms were proposed to explain the association between dietary fat and PCa development and progression, including growth factor signalling (such as IGF-1), inflammation, and endocrine modulation (*Narita et al., 2019*). While the evidence supporting the negative impacts of saturated dietary FAs are more consistent, the effect of dietary PUFAs on PCa aggressiveness remains inconclusive and differences between omega-3 and omega-6 have been reported (*Bairati et al., 1998*; *Stéfani et al., 2000*; *Park et al., 2009*; *Fu et al., 2015*). In contrast to omega-3 PUFAs, which are reported to inhibit PCa progression (*Wang et al., 2012*), omega-6 PUFAs (*Khankari et al., 2016*; *Brown et al., 2010*), and higher omega-6/omega-3 PUFA ratio, increase PCa risk (*Williams et al., 2011*; *Apte et al., 2013*). Dietary intervention by decreasing total fat intake and increasing omega-3 PUFAs was found to improve PCa survivorship (*Epstein et al., 2012*; *Colli and Colli, 2005*; *Davies et al., 2011*; *Ornish et al., 2005*). It is unclear whether there is a preference for n-6 or n-3 PUFA β-oxidation in PCa cells, however both require DECR1 for complete β-oxidation.

Although FAO is a complex process that requires the activity of several enzymes, to date the entire focus of drug development strategies has been inhibitors against CPT1, the rate-limiting step of FA β-oxidation. CPT1 is responsible for synthesizing fatty acyl-carnitines from fatty acyl-CoAs which are then transported from the cytoplasm into the mitochondria by carnitine acylcarnitine translocase for subsequent processing then entry into the β-oxidation pathway. Even though targeting CPT1 is efficient in inhibiting FA β-oxidation, the clinical use of CPT1 inhibitors is challenging. Based on our current findings, we propose that DECR1 is an attractive alternative target to CPT1. CPT1 inhibition would suppress β-oxidation of all long FA species (saturated FA, MUFA and PUFA), whilst in contrast DECR1 is specific for PUFA. Homozygous *CPT1* deficiency, of either the liver or muscle isoform, is lethal in mice (*Nyman et al., 2005*; *Ji et al., 2008*; *Haynie et al., 2014*), but *Decr1*$^{-/-}$ mice are viable, and clinical symptoms arose only after metabolic stress (*Miinalainen et al., 2009*; *Mäkelä et al., 2019*). The marked overexpression of DECR1 in prostate tumors across multiple clinical cohorts, potentially coupled to PUFA-related dietary interventions, may lend further selectivity to targeting strategies. Of note, the crystal structure of DECR1 active site has been solved (*Alphey et al., 2005*), and thus developing DECR1 inhibitors is feasible.

In summary, herein we strengthen the evidence base for the critical importance of FAO and, specifically, PUFA oxidation in PCa, thereby identifying a promising new therapeutic candidate, DECR1.

## Materials and methods

### Meta-Analysis of lipid metabolism genes

Individual RNA-sequencing (RNA-seq) datasets composed of matched normal versus tumor prostate cancer patient tissue samples were acquired and are listed as follows: (*Bray et al., 2018*) The Cancer Genome Atlas (TCGA, n = 53); (*Huggins and Hodges, 1941*) Nikitina AS et al. (GSE89223, n = 10) (*Nikitina et al., 2017*; *Tran et al., 2009*) Ren S et al. (n = 14) (*Ren et al., 2012*; and (*Cai and Balk, 2011*) Ding Y et al. (GSE89194, n = 45) (*Ding et al., 2016*). Before the meta-analysis, RNA-seq data was quality controlled and analysed using the R limma voom-*eBayes pipeline* (*Law et al., 2016*). Effect sizes (log-fold changes) and corresponding variances were collected from the differential expression analysis under the matched-pairs design. Meta-analysis was performed by applying a restricted maximum-likelihood estimator (REML) within a random-effects model using the *rma* function from the R *metafor* package. At most one missing observation out of four was allowed per gene. Next, the retained genes were intersected with the list of pre-selected 735 genes involved in lipid metabolism as identified from REACTOME. Finally, the remaining genes were rank-ordered on the basis of their *meta effect size* scores across all four RNA-seq datasets. Top 20 candidate genes were selected for further disease-free survival association analyses from well-characterized clinical cohorts.

### Clinical datasets

Transcriptomic data was downloaded from The Cancer Genome Atlas (TCGA) data portal, cBioPortal (*Cerami et al., 2012*), and GEO; GSE21032 *Taylor et al., 2010*; GSE35988 *Grasso et al., 2012*; GSE6099 *Tomlins et al., 2007*; GSE16560 (*Sboner et al., 2010*). Proteomics data was extracted from *Iglesias-Gato et al., 2018*.

### Cell lines and tissue culture

The human normal prostate epithelial cell lines PNT1 and PNT2 were obtained from the European Collection of Authenticated Cell Cultures (ECACC), and prostate carcinoma cells LNCaP, VCaP, and 22RV1 were obtained from the American Type Culture Collection (ATCC; Rockville, MD, USA). Castrate resistant V16D and enzalutamide resistant MR49F cells were derived through serial xenograft passage of LNCaP cells (*Toren et al., 2016*) were a kind gift from Prof. Amina Zoubeidi laboratory. Cell lines were verified in 2018 via short tandem repeat profiling (Cell Bank Australia). Cell lines were maintained in RPMI-1640 medium containing 10% fetal bovine serum (FBS; Sigma-Aldrich, NSW, Australia) in 5% $CO_2$ in a humidified atmosphere at 37°C. Prior to androgen treatment, cells were seeded in medium supplemented with 5% dextran charcoal coated FBS (DCC-FBS) and after 24 hr, 1 nM or 10 nM of dihydrotestosterone (DHT) was added. For anti-androgen treatment, cells were cultured in growth medium supplemented with 2.5 µM, 5 µM, 7.5 µM or 10 µM Enzalutamide (dissolved in dimethyl sulfoxide, DMSO; Sigma Aldrich). The sources and experimental conditions for primary antibodies used in this study are listed in the Key Resources Table. Primers were obtained from Sigma-Aldrich and their sequences are detailed in the Key Resources Table.

### Ex vivo culture of human prostate tumors

Patient derived-explant culture was carried out according to techniques established in our laboratory and as described previously (*Armstrong et al., 2016*; *Centenera et al., 2018*; *Centenera et al., 2013*). 6 mm/8 mm biopsy cores were collected from men undergoing robotic radical prostatectomy at St. Andrew's Hospital (Adelaide, South Australia) with written informed consent through the Australian Prostate Cancer BioResource. The tissue was dissected into smaller 1 $mm^3$ pieces and cultured on Gelfoam sponges (80 × 125 mm Pfizer 1205147) in 24-well plates pre-soaked in 500 µl RPMI-1640 with 10% FBS, antibiotic/antimycotic solution. Etomoxir (100 µM) or Enzalutamide (10 µM) was added into each well and the tissues were cultured in 5% $CO_2$ in a humidified atmosphere at 37°C for 48 hr, then snap frozen in liquid nitrogen and stored at −80°C, or formalin-fixed and paraffin-embedded.

## Immunohistochemistry (IHC)

Paraffin-embedded tissue sections (2–4 µm) were deparaffinized in xylene, rehydrated through graded ethanol, and blocked for endogenous peroxidase before being subjected to heat-induced epitope retrieval (*Armstrong et al., 2016*). IHC staining was performed using DECR1 (HPA023238, Sigma Aldrich, diluted 1:500) antibody and the 3,3'-Diaminobenzidine (DAB) Enhanced Liquid Substrate System tetrahydrochloride (Sigma Aldrich) as described previously (*Armstrong et al., 2016*). Intensity of DECR1 immunostaining was measured by video image analysis (*Armstrong et al., 2016*).

## Western blotting

Protein lysates were collected in RIPA lysis buffer (10 mM Tris, 150 mM NaCl, 1 mM EDTA, 1% Triton X-100, 10% protease inhibitor). Western blotting on whole cell protein lysates were performed as previously described (*Armstrong et al., 2016*). *Cell Fractionation*. Protein lysates from each subcellular fraction (cytoplasm, mitochondria, and nucleus) were obtained from PNT1 and LNCaP cells using the cell fractionation kit (Abcam, VIC, Australia) according to the manufacturer's protocol.

## Quantitative Real-Time PCR (qPCR)

RNA was extracted from cells using the RNeasy RNA extraction kit (Qiagen), followed by the iScript cDNA Synthesis kit (Bio-Rad, NSW, Australia). qPCR was performed with a 1:10 dilution of cDNA using SYBR Green (Bio-Rad) on a CFX384 Real-Time System (Bio-Rad). Relative gene expression was calculated using the comparative Ct method and normalized to the internal control genes *GUSB* and *L19* for prostate cancer cells and LNCaP-derived tumors, or *TUBA1B, PPIA* and *GAPDH* for PDEs.

## Analysis of published ChIP-seq data

AR ChIP-seq data from published external datasets, GSE56288 (clinical specimens; seven normal prostate and 13 primary tumors) (*Pomerantz et al., 2015*) and GSE55064 (VCaP cell line; Veh, DHT treated, MDV3100 treated and Bicalutamide treated) (*Asangani et al., 2014*) were obtained from GEO and visualized using the Integrated Genome Browser (IGV).

## Chromatin immunoprecipitation (ChIP)

LNCaP cells were seeded at $3 \times 10^6$ cells/plate in 15 cm plates in RPMI-1640 medium containing 10% DCC-FBS for 3 days, then treated for 4 hr with 10 nM DHT or Vehicle (ethanol). AR ChIP was performed as described previously (*Paltoglou et al., 2017*).

## Transient RNA interference

The human DECR1 ON-TARGET plus small interfering RNAs (siRNAs) and control siRNA (D-001810-01-20 ON-TARGET plus Non-targeting siRNA #1) were purchased from Millennium Science (VIC, Australia). Four siRNA were tested and the two most effective were selected for our experimentation: siDECR1-1 (J-009642-05-0002) and siDECR1-2 (J-009642-06-0002). The siRNAs at a concentration of 5 nM were reverse transfected using Lipofectamine RNAiMAX transfection reagent (Invitrogen, VIC, Australia) according to the manufacturer's protocol. DECR1 downregulation (>80%) was confirmed on mRNA and protein levels.

## Generation of stable shDECR1 and hDECR1 LNCaP cells

### Short hairpin lentiviral expression vector

LNCaP cells were transduced with the universal negative control shRNA lentiviral particles (shControl), DECR1 shRNA lentiviral particles (shDECR1) or hDECR1 (GFP-Puro) designed by GenTarget Inc (San Diego, CA, USA) according to the manufacturer's protocol (*Figure 5—figure supplement 3*).

## Functional assays

### Cell viability

Cells were seeded in triplicates in 24-well plates at a density of $3.0 \times 10^4$–$6.0 \times 10^4$ cells/well and reverse transfected with siRNA overnight. Cells were manually counted using a hemocytometer 96

hr post-siRNA knockdown and viability was assessed by Trypan Blue exclusion as described previously (*Armstrong et al., 2016*).

## Cell migration

Transwell migration assays were performed using 24-well polycarbonate Transwell inserts (3422, Sigma Aldrich). LNCaP, 22RV1 and MR49F cells transfected overnight with siRNA were seeded into the upper chamber of the Transwell at a density of $8 \times 10^4$–$2.5 \times 10^5$ cells/well in serum-free medium. 650 µl of medium containing 5% FBS was added to the bottom chamber. Cells were incubated at 37°C for 48 hr. Non-migrated cells were gently removed using a cotton-tipped swab. The inserts were fixed in 4% paraformaldehyde for 20 min and stained with 1% crystal violet for 30 min. The images of migrated cells were captured using the Axio Scope A1 Fluorescent Microscope (Zeiss) at 40X magnification. The number of migrated cells were counted manually and presented as percentages relative to control cells ± SEM.

## Cell invasion

Cell invasion were assessed using 24-well-plate BD Biocoat Matrigel Invasion Chambers (In Vitro Technologies, NSW, Australia) according to the supplier instructions. After 48 hr of siRNA transfection, 650 µl of medium containing 10% FBS was added to the bottom chamber, and equal number of cells within 1% FBS-contained medium were transferred to the upper chamber. After incubation at 37°C, 5% $CO_2$ for 48 hr, non-invading cells as well as the Matrigel from the interior of the inserts were gently removed using a cotton-tipped swab. The inserts were fixed in 4% paraformaldehyde for 20 min and stained with 1% crystal violet for 30 min. The images of invaded cells were captured using the Axio Scope A1 Fluorescent Microscope (Zeiss) at 40X magnification. The number of invaded cells were counted manually and presented as percentages relative to control cells ± SEM.

## Colony formation assay

DECR1 stable knockdown cells (shDECR1) or negative control cells (shControl) were prepared in a single-cell suspension before being plated in 6-well plates (500 cells/well). Cells were incubated for 2 weeks at 37°C and medium was replenished every 3–7 days. After 3 weeks, cells were washed with PBS, fixed with 4% paraformaldehyde and stained with 1% crystal violet for 30 min. Colonies were counted manually, and results were reported as number of colonies ± SEM.

## 3D Spheroid growth assay

LNCaP and 22RV1 cells were transfected with siRNA in 6-well plates for 48 hr. Cells were collected and prepared at a concentration of $7.5 \times 10^4$ cells/ml. Cell suspensions (1500 cells in 20 µl) were pipetted onto the inside of a petri dish lid, and 15 ml of PBS was added to the dish to prevent the drops from drying. The petri dishes were reassembled and incubated at 37°C for 5 days. Photos of the formed spheres were captured, and the sphere volume was determined using ReViSP software (*Piccinini et al., 2015*).

## Seahorse extracellular flux analysis

Cells were plated on XF96 well cell culture microplates (Agilent, VIC, Australia) at equal densities in substrate-limited medium (DMEM with 0.5 mM glucose, 1.0 mM glutamine, 0.5 mM carnitine and 1% FBS) and incubated overnight. One hour before the beginning of OCR measurement, the cells were changed into FAO Assay Medium (111 mM NaCl, 4.7 mM KCl, 2.0 mM $MgSO_4$, 1.2 mM $Na_2HPO_4$, 2.5 mM glucose, 0.5 mM carnitine and 5 mM HEPES). After baseline OCR is stabilized in FAO Assay Medium, 200 µM of linoleic-acid (LA) or palmitic acid (PA) were added before initializing measurements. Extracellular flux analysis was performed using the Seahorse XF Cell Mitochondrial Stress Test kit (Seahorse Bioscience) according to the manufacturer's protocol. Extracellular flux experiments were performed on a Seahorse XF96 Analyzer and results were analysed using Seahorse Wave software for XF analyzers. The OCR values were normalized to cell numbers in each well.

## Metabolomics

LNCaP cells were transfected with siRNA for 96 hr in no glucose medium (containing 10% FBS) supplemented with 2.5 mM glucose in 6-well plates. Cells were washed with 1 ml of 37°C Milli-Q water

on the shaker for 2 s. The plate was placed in sufficient volumes of liquid nitrogen, enough to cover the surface of the plate and was briefly stored on dry ice. 600 µl of ice cold 90% 9:1 methanol:chloroform ($MeOH:CHCl_3$) extraction solvent containing the internal standards (0.5 µl/samples) was added onto each well and allowed to incubate for 10 min. Cells were collected into 1.5 ml Eppendorf tubes, incubated on ice for 5 min, and centrifuged at 4°C for 5 min at 16,100 g. The supernatant was then transferred into a fresh 1.5 ml Eppendorf tube and allowed to dry in a Speedvac. Dried samples were derivatised with 20 µl methoxyamine (30 mg/ml in pyridine, Sigma Aldrich) and 20 µl N,O-Bis(trimethylsilyl)trifluoroacetamide (BSTFA) + 1% Trimethylchlorosilane (TMCS). The derivatised samples were analysed using GC QQQ targeted metabolomics as described in *Best et al., 2018*.

## Lipidomics

### Lipid extraction

700 µl of sample (4 µl of plasma diluted in water, or 700 µl of homogenized cells) was mixed with 800 µl 1 N HCl:CH3OH 1:8 (v/v), 900 µl CHCl3 and 200 µg/ml of the antioxidant 2,6-di-tert-butyl-4-methylphenol (BHT; Sigma Aldrich). 3 µl of SPLASH LIPIDOMIX Mass Spec Standard (#330707, Avanti Polar Lipids) was spiked into the extract mix. The organic fraction was evaporated using a Savant Speedvac spd111v (Thermo Fisher Scientific) at room temperature and the remaining lipid pellet was stored at - 20°C under argon.

### Mass spectrometry

Lipid pellets were reconstituted in 100% ethanol. Lipid species were analyzed by liquid chromatography electrospray ionization tandem mass spectrometry (LC-ESI/MS/MS) on a Nexera X2 UHPLC system (Shimadzu) coupled with hybrid triple quadrupole/linear ion trap mass spectrometer (6500+ QTRAP system; AB SCIEX). Chromatographic separation was performed on a XBridge amide column (150 mm ×4.6 mm, 3.5 µm; Waters) maintained at 35°C using mobile phase A [1 mM ammonium acetate in water-acetonitrile 5:95 (v/v)] and mobile phase B [1 mM ammonium acetate in water-acetonitrile 50:50 (v/v)] in the following gradient: (0–6 min: 0% B → 6% B; 6–10 min: 6% B → 25% B; 10–11 min: 25% B → 98% B; 11–13 min: 98% B → 100% B; 13–19 min: 100% B; 19–24 min: 0% B) at a flow rate of 0.7 mL/min which was increased to 1.5 mL/min from 13 min onwards. SM, CE, CER, DCER, HCER, LCER were measured in positive ion mode with a precursor scan of 184.1, 369.4, 264.4, 266.4, 264.4 and 264.4 respectively. TAG, DAG and MAG were measured in positive ion mode with a neutral loss scan for one of the fatty acyl moieties. PC, LPC, PE, LPE, PG, LPG, PI, LPI, PS and LPS were measured in negative ion mode by fatty acyl fragment ions. Lipid quantification was performed by scheduled multiple reactions monitoring (MRM), the transitions being based on the neutral losses or the typical product ions as described above. The instrument parameters were as follows: Curtain Gas = 35 psi; Collision Gas = 8 a.u. (medium); IonSpray Voltage = 5500 V and −4,500 V; Temperature = 550°C; Ion Source Gas 1 = 50 psi; Ion Source Gas 2 = 60 psi; Declustering Potential = 60 V and −80 V; Entrance Potential = 10 V and −10 V; Collision Cell Exit Potential = 15 V and −15 V. The following fatty acyl moieties were taken into account for the lipidomic analysis: 14:0, 14:1, 16:0, 16:1, 16:2, 18:0, 18:1, 18:2, 18:3, 20:0, 20:1, 20:2, 20:3, 20:4, 20:5, 22:0, 22:1, 22:2, 22:4, 22:5 and 22:6 except for TGs which considered: 16:0, 16:1, 18:0, 18:1, 18:2, 18:3, 20:3, 20:4, 20:5, 22:2, 22:3, 22:4, 22:5, 22:6.

### Data analysis

Peak integration was performed with the MultiQuant software version 3.0.3. Lipid species signals were corrected for isotopic contributions (calculated with Python Molmass 2019.1.1) and were normalized to internal standard signals. Unpaired T-test p-values and FDR corrected p-values (using the Benjamini/Hochberg procedure) were calculated in Python StatsModels version 0.10.1.

## Mitochondrial ROS measurement

LNCaP cells were transfected with siRNA for 96 hr in 6-well plates. Cells were collected into fluorescence-activated cell sorting (FACS) tubes and stained with 2.5 µM of MitoSOX Red stain (Thermo Fisher Scientific, VIC, Australia) for 30 min in a 37°C water bath. Cells were centrifuged at 1,500 rpm for 5 min, washed twice with 500 µl of PBS, and resuspended in 500 µl of pre-warmed PBS before the samples are read on a BD FACSymphony flow cytometer.

## Acylcarnitine measurement

Total lipids were extracted from cells using two-phase extraction with methyl-tert-butyl-ether (MTBE)/methanol/water (10:3:2.5, v/v/v) (*Matyash et al., 2008*). Cell pellets were frozen in 8:1 methanol/water prior to extraction with the above solvent mixture; for cell culture supernatant samples, the cell culture medium replaced the water component. Deuterated (D3)-palmitoylcarnitine was included as an internal standard (200 pmole/sample for LNCaP samples; 20 pmole/sample for tumor explant samples). Samples were reconstituted in 200 µL of the HPLC starting condition, defined below.

Acylcarnitines were quantified by liquid chromatography-tandem mass spectrometry using a Q-Exactive HF-X mass spectrometer with heated electrospray ionization and a Vantage HPLC system (ThermoFisher Scientific). Extracts were resolved on a 2.1 × 100 mm Waters Acquity C18 UPLC column (1.7 µm pore size), using an 18 min binary gradient at 0.28 mL/min flow rate, as follows: 0 min, 80:20 A/B; 3 min, 80:20 A/B; 6 min, 57:43 A/B; 8 min, 35:65 A/B; 9 min, 0:100 A/B; 14 min, 0:100 A/B; 14.5 min, 80:20 A/B; 18 min, 80:20 A/B. Solvent A: 10 mM ammonium formate, 0.1% formic acid in acetonitrile:water (60:40); Solvent B: 10 mM ammonium formate, 0.1% formic acid in isopropanol:acetonitrile (90:10). Data was acquired in positive ion mode with data-dependent acquisition (full scan resolution 70,000 FWHM, scan range 220–1600 $m/z$). The ten most abundant ions in each cycle were subjected to fragmentation (collision energy 30 eV, resolution 17,500 FWHM). An exclusion list of background ions was used based on a solvent blank. TraceFinder v5.0 (Thermo Fisher Scientific) was used for peak detection and integration, based on exact precursor ion mass ($m/z$ tolerance four ppm) and $m/z$ 85.0 acylcarnitine product ion. Peak areas were normalised to the D3-palmitoylcarnitine internal standard.

## Lipid peroxidation analysis by imaging

For imaging, LNCaP cells following DECR1 knockdown were plated at $5 \times 10^3$ cells/well in a 8-well chamber slide. Cells were then washed with Hank's balanced salt solution (HBSS) and incubated with 5 µM BODIPY-581/591 C11 stain (Thermo Fisher Scientific). Cells were washed and fixed with 4% paraformaldehyde (PFA), and mounted with Prolong Gold anti-fade solution with DAPI (Thermo Fisher Scientific). Cells were imaged at 60 X magnification using a Olympus FV3000 Confocal Microscope. Quantification of BODIPY-C11 stain was performed using ImageJ analysis software.

## In vivo experiments

### Castration + ENZ study

LNCaP cells ($5 \times 10^6$ cells in 50 µL 10% FBS/RPMI 1640 medium) were co-injected subcutaneously with 50 µL Matrigel in 6-week-old NOD Scid Gamma male mice (Bioresource Facility, Austin Health, Heidelberg, Australia). When tumors reached ~200 mm³, mice were randomized in different therapy groups. One group was left untreated ($n = 5$), one group was treated with vehicle control (10% DMSO/PBS; $n = 5$), one group was treated with enzalutamide (10 mg/kg MDV3100 in 10% DMSO/PBS) and one group was castrated by surgical castration under isofluorane anesthesia ($n = 9$). Five of the ten castrated mice were then treated daily with enzalutamide (10 mg/kg MDV3100 in 10% DMSO/PBS) by oral gavage for 7 days. Enzalutamide therapy of castrated mice started five days after surgery.

### Subcutaneous tumor growth

DECR1 stable knockdown cells (shDECR1) or negative control cells (shControl) ($5 \times 10^6$ cells in 50 µL 10% FBS/RPMI 1640 medium) were co-injected subcutaneously with 50 µL matrigel in 6-week-old NOD Scid Gamma male mice. Tumors were measured using callipers and their volumes were calculated using the formula length × width²/2.

### Orthotopic tumor growth

Ten microliter containing $1 \times 10^6$ DECR1 stable knockdown cells (shDECR1) or negative control cells (shControl) were -injected intraprostatically in 8 week old NOD/SCID male mice. Whole-body imaging to monitor luciferase-expressing LNCaP cells was performed at day 3 of the injection and once weekly after that using the IVIS Spectrum In Vivo Imaging System (PerkinElmer). D-luciferin (potassium salt, PerkinElmer) was dissolved in sterile deionized water (0.03 g/ml) and injected

subcutaneously (3 mg/20 g of mouse body weight) before imaging. Bioluminescence is reported as the sum of detected photons per second from a constant region of interest. After the animals were sacrificed, lungs and livers were excised for ex vivo imaging using the IVIS system.

After each study, tumors were excised and half was snap frozen for RNA extraction while the other half was formalin fixed and paraffin embedded. All animal procedures were carried out in accordance with the guidelines of the National Health and Medical Research Council of Australia, with subcutaneous xenograft studies approved by the Austin Health Animal Ethics Committee (approval number A2015/05311) and orthotopic xenograft studies approved by the University of Adelaide Animal Ethics Committee (approval number M-2019–037).

## Statistical analysis

Results are reported as mean ± S.E.M. Statistical analysis was performed using GraphPad Prism (V7.0 for Windows). Differences between treatment groups were compared by T-test or one-way ANOVA followed by Tukey or Dunnett post hoc test. Significance is expressed as $*p < 0.05$, $**p < 0.01$, $***p < 0.001$, $****p < 0.0001$.

## Acknowledgements

The results published here are in part based on data generated by The Cancer Genome Atlas, established by the National Cancer Institute and the National Human Genome Research Institute, and we are grateful to the specimen donors and relevant research groups associated with this project. Tissues for the patient-derived explants used in the study were collected with informed consent via the Australian Prostate Cancer BioResource and we thank the doctors, patients and health care professionals involved. We acknowledge expert technical assistance in the study from Natalie Ryan, Joanna Gillis, Kayla Bremert, Samira Khabbazi, Nhi Huynh, and Holly P McEwen. Flow cytometry analysis was performed at the South Australian Health Medical Research Institute (SAHMRI) in the ACRF Cellular Imaging and Cytometry Core Facility. The Facility is generously supported by the Australian Cancer Research Foundation (ACRF), Detmold Hoopman Group and Australian Government through the Zero Childhood Cancer Program. Animal studies were performed at the Bioresource Facilities at Austin Health, Heidelberg, Australia and the South Australian Health and Medical Research Institute. The authors thank Metabolomics Australia, Bio21 Institute, and Adelaide Microscopy (University of Adelaide).

## Additional information

### Funding

| Funder | Grant reference number | Author |
|---|---|---|
| National Health and Medical Research Council | 1138648 | Zeyad D Nassar |
| National Health and Medical Research Council | 1121057 | Luke A Selth |
| National Health and Medical Research Council | 1100626 | Anthony S Don |
| National Health and Medical Research Council | 1084178 | Andrew M Scott |
| Prostate Cancer Foundation of Australia | YI 1417 | Zeyad D Nassar |
| Cure Cancer Australia Foundation | 1164798 | Zeyad D Nassar |
| EMBL Australia | Group Leader Award | David J Lynn |
| University of Sydney | Robinson Fellowship | Andrew J Hoy |
| Fonds Wetenschappelijk Onderzoek | Project Grants G.0841.15 and G.0C22.19N | Johannes V Swinnen |

| KU Leuven | Project Grants C16/15/073 and C32/17/052 | Johannes V Swinnen |
|---|---|---|
| Australian Research Council | FT130101004 | Lisa M Butler |
| Cancer Council South Australia | PRF1117 | Lisa M Butler |
| Movember Foundation | MRTA3 | Lisa M Butler |
| Freemasons Foundation Centre for Men's Health, University of Adelaide | | Lisa M Butler |

The funders had no role in study design, data collection and interpretation, or the decision to submit the work for publication.

## Author contributions

Zeyad D Nassar, Conceptualization, Data curation, Formal analysis, Supervision, Funding acquisition, Investigation, Methodology, Writing - original draft, Writing - review and editing; Chui Yan Mah, Data curation, Formal analysis, Validation, Investigation, Visualization, Methodology, Writing - original draft, Writing - review and editing; Jonas Dehairs, Data curation, Methodology, Writing - review and editing; Ingrid JG Burvenich, Swati Irani, Anthony S Don, Investigation, Methodology, Writing - review and editing; Margaret M Centenera, Supervision, Methodology, Writing - review and editing; Madison Helm, Investigation, Methodology; Raj K Shrestha, Investigation, Writing - review and editing; Max Moldovan, Formal analysis, Investigation, Writing - review and editing; Jeff Holst, Lisa G Horvath, Methodology, Writing - review and editing; Andrew M Scott, Supervision, Writing - review and editing; David J Lynn, Data curation, Formal analysis, Supervision, Methodology, Writing - review and editing; Luke A Selth, Data curation, Formal analysis, Supervision, Investigation, Visualization, Methodology, Writing - review and editing; Andrew J Hoy, Data curation, Supervision, Investigation, Methodology, Writing - review and editing; Johannes V Swinnen, Conceptualization, Data curation, Supervision, Funding acquisition, Methodology, Writing - review and editing; Lisa M Butler, Conceptualization, Resources, Data curation, Supervision, Funding acquisition, Writing - original draft, Project administration, Writing - review and editing

## Author ORCIDs

Zeyad D Nassar (iD) http://orcid.org/0000-0002-7779-2697
Chui Yan Mah (iD) http://orcid.org/0000-0002-8820-4037
Lisa M Butler (iD) https://orcid.org/0000-0003-2698-3220

## Ethics

Human subjects: Fresh and archival prostate tissue specimens were collected from men undergoing robotic radical prostatectomy at St Andrew's Hospital (Adelaide, South Australia) with written informed consent through the Australian Prostate Cancer BioResource. Ethical Approval was provided by the Human Research Ethics Committees of the University of Adelaide (H-2012-016) and St Andrew's Hospital.

Animal experimentation: Animal studies were approved by the Austin Health Animal Ethics Committee (approval number A2015/05311), Heidelberg, Australia, and the University of Adelaide Animal Ethics Committee (approval number M-2019-037), and were carried out in accordance with the recommendations of the National Health and Medical Research Council of Australia.

## Decision letter and Author response

Decision letter https://doi.org/10.7554/eLife.54166.sa1
Author response https://doi.org/10.7554/eLife.54166.sa2

# Additional files

## Supplementary files

- Transparent reporting form

## Data availability

All data generated or analysed during this study are included in the manuscript and supporting files.

The following previously published datasets were used:

| Author(s) | Year | Dataset title | Dataset URL | Database and Identifier |
|---|---|---|---|---|
| Nikitina AS, Sharova EI, Danilenko SA, Butusova TB, Vasiliev AO, Govorov AV, Prilepskaya EA, Pushkar DY, Kostryukova ES | 2017 | Novel RNA biomarkers of prostate cancer revealed by RNA-seq analysis of formalin-fixed samples obtained from Russian patients | https://www.ncbi.nlm.nih.gov/query/acc.cgi?acc=GSE89223 | NCBI Gene Expression Omnibus, GSE89223 |
| Ding Y, Wu H, Warden C, Steele L, Liu X, Iterson MV, Wu X, Nelson R, Liu Z, Yuan YC, Neuhausen SL | 2016 | Age-Related Gene Expression Changes in Prostate Cancer Patients | https://www.ncbi.nlm.nih.gov/geo/query/acc.cgi?acc=GSE89194 | NCBI Gene Expression Omnibus, GSE89194 |
| Ren S, Peng Z, Mao JH, Yu Y, Yin C, Gao X, Cui Z, Zhang J, Yi K, Xu W, Chen C, Wang F, Guo X, Lu J, Yang J, Wei M, Tian Z, Guan Y, Tang L, Xu C, Wang L, Tian W, Wang J, Yang H, Sun Y | 2012 | RNA-seq analysis of prostate cancer in the Chinese population identifies recurrent gene fusions, cancer-associated long noncoding RNAs and aberrant alternative splicings | https://www.ncbi.nlm.nih.gov/pubmed/22349460 | Pubmed, 22349460 |
| TCGA | 2019 | The Cancer Genome Atlas Prostate Adenocarcinoma (TCGA-PRAD) | https://portal.gdc.cancer.gov/projects/TCGA-PRAD | National Cancer Institute, TCGA-PRAD |
| Tomlins SA, Mehra R, Rhodes DR, Cao X, Wang L, Dhanasekaran SM, Kalyana-Sundaram S, Wei JT, Rubin MA, Pienta KJ, Shah RB, Chinnaiyan AM | 2006 | Integrative Molecular Concepts Modeling of Prostate Cancer Progression | https://www.ncbi.nlm.nih.gov/geo/query/acc.cgi?acc=GSE6099 | NCBI Gene Expression Omnibus, GSE6099 |
| Grasso CS, Wu YM, Robinson DR, Cao X, Dhanasekaran SM, Khan AP, Quist MJ, Jing X, Lonigro RJ, Brenner JC, Asangani IA, Ateeq B, Chun SY, Siddiqui J, Sam L, Anstett M, Mehra R, Prensner JR, Palanisamy N, Ryslik GA, Vandin F, Raphael BJ, Kunju LP, Rhodes DR, Pienta KJ, Chinnaiyan AM, Tomlins SA | 2012 | The Mutational Landscape of Lethal Castrate Resistant Prostate Cancer | https://www.ncbi.nlm.nih.gov/geo/query/acc.cgi?acc=GSE35988 | NCBI Gene Expression Omnibus, GSE35988 |
| Taylor BS, Schultz N, Hieronymus H, Gopalan A, Xiao Y, Carver BS, Arora VK, Kaushik P, Cerami E, Reva B, Antipin Y, Mitsiades N, Landers T, Dolgalev I, Major JE, Wilson M, Socci | 2010 | Integrative genomic profiling of human prostate cancer | https://www.ncbi.nlm.nih.gov/geo/query/acc.cgi?acc=GSE21032 | NCBI Gene Expression Omnibus, GSE21032 |

| | | | | |
|---|---|---|---|---|
| ND, Lash AE, Heguy A, Eastham JA, Scher HI, Reuter VE, Scardino PT, Sander C, Sawyers CL, Gerald WL | | | | |
| Sboner A, Demichelis F, Calza S, Pawitan Y, Setlur SR, Hoshida Y, Perner S, Adami HO, Fall K, Mucci LA, Kantoff PW, Stampfer M, Andersson SO, Varenhorst E, Johansson JE, Gerstein MB, Golub TR, Rubin MA, Andrén O | 2010 | Molecular Sampling of Prostate Cancer: a dilemma for predicting disease progression | https://www.ncbi.nlm.nih.gov/geo/query/acc.cgi?acc=GSE16560 | NCBI Gene Expression Omnibus, GSE16560 |
| Wang Q, Li W, Liu XS, Carroll JS, Jänne OA, Keeton EK, Chinnaiyan AM, Pienta KJ, Brown M | 2007 | A hierarchical network of transcription factors governs androgen receptor-dependent prostate cancer growth | https://www.ncbi.nlm.nih.gov/geo/query/acc.cgi?acc=GSE7868 | NCBI Gene Expression Omnibus, GSE7868 |
| Heemers HV, Schmidt LJ, Sun Z, Regan KM, Anderson SK, Duncan K, Wang D, Liu S, Ballman KV, Tindall DJ | 2011 | Identification of an SRF- and androgen-dependent gene signature in prostate cancer | https://www.ncbi.nlm.nih.gov/geo/query/acc.cgi | NCBI Gene Expression Omnibus, GSE22606 |
| Asangani IA, Dommeti VL, Wang X, Malik R, Cieslik M, Yang R, Escara-Wilke J, Wilder-Romans K, Dhaniredddy S, Engelke C, Iyer MK, Jing X, Wu YM, Cao X, Qin ZS, Wang S, Feng FY, Chinnaiyan AM | 2014 | Therapeutic Targeting of BET Bromodomain Proteins in Castration-Resistant Prostate Cancer | https://www.ncbi.nlm.nih.gov/geo/query/acc.cgi?acc=GSE55064 | NCBI Gene Expression Omnibus, GSE55064 |
| Li L, Karanika S, Yang G, Wang J, Park S, Broom BM, Manyam GC, Wu W, Luo Y, Basourakos S, Song JH, Gallick GE, Karantanos T, Korentzelos D, Azad AK, Kim J, Corn PG, Aparicio AM, Logothetis CJ, Troncoso P, Heffernan T, Toniatti C, Lee HS, Lee JS, Zuo X, Chang W, Yin J, Thompson TC | 2016 | Genome-wide analysis of enzalutamide- and/or olaparib-responsive gene expression in prostate cancer cells | https://www.ncbi.nlm.nih.gov/geo/query/acc.cgi?acc=GSE69249 | NCBI Gene Expression Omnibus, GSE69249 |
| Wang X, Wang B, Soriano R, Zha J, Zhang Z, Modrusan Z, Cunha GR, Gao W | 2006 | Expression profiling of the mouse prostate after castration and hormone replacement | https://www.ncbi.nlm.nih.gov/geo/query/acc.cgi?acc=GSE5901 | NCBI Gene Expression Omnibus, GSE5901 |
| Arora VK, Schenkein E, Murali R, Subudhi SK, Wongvipat J, Balbas MD, Shah N, Cai L, Efstathiou E, | 2013 | Glucocorticoid Receptor Confers Resistance to Anti-Androgens by Bypassing Androgen Receptor Blockade | https://www.ncbi.nlm.nih.gov/geo/query/acc.cgi?acc=GSE52169 | NCBI Gene Expression Omnibus, GSE52169 |

| | | | | |
|---|---|---|---|---|
| Logothetis C, Zheng D, Sawyers CL | | | | |
| Pomerantz MM, Li F, Takeda D, Chonkar A, Chabot M, Li Q, Cejas P, Vazquez F, Shivda-sani RA, Seo J, Bowden M, Lis R, Hahn WC, Kantoff PW, Brown M, Loda M, Long HW, Freedman ML | 2015 | Androgen receptor programming in human tissue implicates HOXB13 in prostate pathogenesis [ChIP-Seq] | https://www.ncbi.nlm.nih.gov/geo/query/acc.cgi?acc=GSE56288 | NCBI Gene Expression Omnibus, GSE56288 |
| Glinsky GV, Glinskii AB, Stephenson AJ, Hoffman RM, Gerald WL | 2004 | Gene expression profiling predicts clinical outcome of prostate cancer | https://www.ncbi.nlm.nih.gov/pubmed/15067324 | Pubmed, 150 67324 |
| Latonen L, Afyou-nian E, Jylhä A, Nättinen J, Aapola U, Annala M, Kivi-nummi KK, Tam-mela TTL, Beuerman RW, Uu-sitalo H, Nykter M, Visakorpi T | 2018 | Integrative proteomics in prostate cancer uncovers robustness against genomic and transcriptomic aberrations during disease progression | https://www.nature.com/articles/s41467-018-03573-6#Sec31 | Peptide Atlas repository, PASS0 1126 |
| Iglesias-Gato D, Thysell E, Tyanova S, Crnalic S, Santos A, Lima TS, Geiger T, Cox J, Widmark A, Bergh A, Mann M, Flores-Morales A, Wikström P | 2018 | The Proteome of Prostate Cancer Bone Metastasis Reveals Heterogeneity with Prognostic Implications | https://clincancerres.aacrjournals.org/content/24/21/5433 | ProteomeXchange, PXD009868 |

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

## Appendix 1

# Key Resources Table

**Appendix 1—key resources table**

| Reagent type (species) or resource | Designation | Source or reference | Identifiers | Additional information |
|---|---|---|---|---|
| Strain, strain background (*M. musculus*, male) | NOD scid Gamma (*M. musculus*, male,) Mice | The Jackson Laboratory/Interbred at SAHMRI Biore-sources | NOD.Cg-Prkdcscid/J RRID:IMSR_JAX:001303 | |
| Cell line (*Homo-sapiens*) | LNCaP | ATCC | ATCC CRL-1740 RRID:CVCL_1379 | |
| Cell line (*Homo-sapiens*) | VCaP | ATCC | ATCC CRL-2876 RRID:CVCL_2235 | |
| Cell line (*Homo-sapiens*) | 22RV1 | ATCC | ATCC CRL-2505 RRID:CVCL_1045 | |
| Cell line (*Homo-sapiens*) | PNT1A | The European Collection of Authenticated Cell Cultures (ECACC) | Cat# 95012614 RRID:CVCL_2163 | |
| Cell line (*Homo-sapiens*) | PNT2 | The European Collection of Authenticated Cell Cultures (ECACC) | Cat# 95012613 RRID:CVCL_2164 | |
| Cell line (*Homo-sapiens*) | V16D | PMID:27046225 | Kind gift from Prof. Amina Zoubeidi | |
| Cell line (*Homo-sapiens*) | MR49F | PMID:27046225 | Kind gift from Prof. Amina Zoubeidi | |
| Transfected construct (*Homo sapiens*) | control siRNA | Dharmacon | D-001810-01-20 ON-TAR-GET plus Non-target-ing siRNA #1 | transfected construct (human) |
| Transfected construct (*Homo sapiens*) | siDECR1-1 | Dharmacon | J-009642-05-0002 | transfected construct (human) |
| Transfected construct (*Homo sapiens*) | siDECR1-2 | Dharmacon | J-009642-06-0002 | transfected construct (human) |
| Transfected construct (*Homo sapiens*) | DECR1 shRNA lenti-vector | GenTargrt | LVS-1002 | Lentiviral construct to transfect and express the shRNA. |

*Appendix 1—key resources table continued*

| Reagent type (species) or resource | Designation | Source or reference | Identifiers | Additional information |
|---|---|---|---|---|
| Transfected construct (*Homo sapiens*) | hDECR1 Overexpressing Lentivector | GenTargrt | LVS-2002 | Lentiviral construct to transfect and over-express DECR1. |
| Transfected construct (*Homo sapiens*) | Negative control shRNA lentivector | GenTargrt | LVS-1002 | Lentiviral construct to transfect and express the shRNA. |
| Antibody | Anti-human $\beta$-Actin (Mouse monoclonal) | Sigma-Aldrich | Cat#: A5441 RRID:AB_476744 | (WB 1:2000) |
| Antibody | Anti-human-HSP90 (Rabbit Polyclonal) | Cell Signalling Technology | Cat#: 48745 RRID:CVCL_E547 | (WB 1:1000) |
| Antibody | Anti-human DECR1 (Rabbit Polyclonal) | Prestige Antibodies (Sigma-Aldrich) | Cat#: HPA023238 RRID:AB_1847587 | (WB 1:1000) (IHC: 1:500) |
| Antibody | Anti-human Malondialdehyde (Rabbit Polyclonal) | Abcam | Cat#: ab6463 RRID:AB_305484 | (WB 1:1000) |
| Antibody | Anti- human Androgen receptor (Rabbit Polyclonal) | Santa Cruz Biotechnology | Cat#: sc-816 RRID:AB_1563391 | (WB 1:1000) |
| Antibody | Anti-human PARP (Rabbit Polyclonal) | Cell Signalling Technology | Cat#: 9542 RRID:AB_592473 | (WB 1:1000) |
| Antibody | Anti-human Cyto-chrome C (Rabbit Polyclonal) | Abcam | Cat#: ab90529 RRID:AB_10673869 | (WB 1:2000) |
| Other | MitoTracker Red CMXRos | Thermo Fisher Scientific | Cat#: M7512 | ICC 1:1000 |
| Other | MitoSOX Red Mitochondrial Superoxide Indicator | Thermo Fisher Scientific | Cat#: M36008 | Flow Cytometry: 2.5 μM |
| Other | 3,3'-Diaminobenzi-dine (DAB) Enhanced Liquid Substrate System tetrahydrochloride | Sigma Aldrich | Cat#: D3939 | |
| Other | BODIPY-C11 | Thermo Fisher Scientific | Cat#: D3861 | Imaging: 5 μM |
| Antibody | Anti-human KI67 (Mouse monoclonal) | DAKO | Cat#: M7240 RRID:AB_2142367 | (IHC 1:200) |
| Antibody | Anti-human AR (Rabbit polyclonal) | Santa Cruz | Cat#: sc-816 RRID:AB_1563391 | (WB 1:1000) |
| Sequence-based reagent | DECR1_F | This paper | PCR primers | CTAAATGGCA-CAGCCTTCGT |

*Appendix 1—key resources table continued*

| Reagent type (species) or resource | Designation | Source or reference | Identifiers | Additional information |
|---|---|---|---|---|
| Sequenced-based reagent | DECR1_R | This paper | PCR primers | AACCTGAACCAG TCTCAGCA |
| Sequence-based reagent | GAPDH_F | This paper | PCR primers | TGCACCACCAAC TGCTTAGC |
| Sequenced-based reagent | GAPDH_R | This paper | PCR primers | GGCATGGACTG TGGTCATGAG |
| Sequence-based reagent | PPIA_F | This paper | PCR primers | GCATACGGGTCC TGGCAT |
| Sequence-based reagent | PPIA_R | This paper | PCR primers | ACATGCTTGCCA TCCAACC |
| Sequence-based re-agent | TUBA1B _F | This paper | PCR primers | CCTTCGCCTCC TAATCCCTA |
| Sequence-based reagent | TUBA1B _R | This paper | PCR primers | CCGTGTTCCAGG-CAGTAGA |
| Sequence-based reagent | MKI67_F | This paper | PCR primers | GCCTGC TCGACCCTACA-GA |
| Sequence-based reagent | MIK67_R | This paper | PCR primers | GCTTGTCAAC TGCGGTTGC |
| Sequence-based reagent | L19_F | This paper | PCR primers | TGCCAG TGGAAAAA TCAGCCA |
| Sequence-based reagent | L19_R | This paper | PCR primers | CAAAGCAAATC TCGACACCTTG |
| Sequence-based reagent | GUSB_F | This paper | PCR primers | CGTCCCACCTA-GAATCTGCT |
| Sequence-based reagent | GUSB_R | This paper | PCR primers | TTGCTCA-CAAAGGTCA-CAGG |
| Sequence-based reagent | DECR1_F | This paper | ChIP-qPCR | TTCTGGAGCGC TAAGAGAGC |
| Sequence-based reagent | DECR1_R | This paper | ChIP-qPCR | AGGGCTTCATC TGACAGTGG |
| Sequence-based reagent | KLK3_F | This paper | ChIP-qPCR | GCCTGGATCTGA-GAGAGATATCA TC |
| Sequence-based reagent | KLK3_R | This paper | ChIP-qPCR | ACACC TTTTTTTTTCTGGA TTGTTG |
| Sequence-based reagent | NC2_F | This paper | ChIP-qPCR | GTGAGTGCCCAG TTAGAGCATCTA |

*Appendix 1—key resources table continued*

| Reagent type (species) or resource | Designation | Source or reference | Identifiers | Additional information |
|---|---|---|---|---|
| Sequence-based reagent | NC2_R | This paper | ChIP-qPCR | GGAACCAGTGGG TCTTGAAGTG |
| Chemical compound, drug | Etomoxir | Sigma Aldrich | Cat#: E1905 | |
| Chemical compound, drug | Dihydrotestosterone | Sigma Aldrich | Cas#: 521-18-6 | |
| Chemical compound, drug | Enzalutamide | Sapphire Bioscience | Cat#: S1250 | |
| Chemical compound, drug | Bovine-serum albumin | Bovostar | Cat#: BSAS-AU | |
| Chemical compound, drug | Linoleic acid | Sigma Aldrich | Cat#: L1376 | |
| Chemical compound, drug | Palmitic acid | Sigma Aldrich | Cat#: P0500 | |
| Chemical compound, drug | D-Luciferin | PerkinElmer | Cat#: 122799 | 3 mg/20 g |
| Chemical compound, drug | Deferoxamine | Sigma Aldrich | Cat#: D9533 | |
| Chemical compound, drug | Ferrostatin | Sigma Aldrich | Cat#: SML0583 | |
| Chemical compound, drug | Erastian | Sigma Aldrich | Cat#: E7781 | |
| Chemical compound, drug | ML210 | Tocris Bioscience | Cat#: 6429 | |
| Chemical compound, drug | FIN56 | Tocris Bioscience | Cat#: 6280 | |
| Chemical compound, drug | cell fractionation kit | Abcam | Cat#: ab109719 | |
| Chemical compound, drug | RNeasy RNA extraction kit | Qiagen | Cat#: 74136 | |
| Chemical compound, drug | iScript cDNA Synthesis kit | Bio-Rad | Cat#: 1708890 | |
| Chemical compound, drug | Seahorse XF Cell Mito chondrial Stress Test kit | Agilent | Cat#: 103015–100 | |
| Software, algorithm | GraphPad Prism | GraphPad Software, Inc | Prism V7 RRID:SCR_ 002798 | |

*Appendix 1—key resources table continued*

| Reagent type (species) or resource | Designation | Source or reference | Identifiers | Additional information |
|---|---|---|---|---|
| Software, algorithm | R | *R Development Core Team, 2019* | R version 3.6.2 RRID:SCR_001905 | |
| Software, algorithm | ReViSP | PMID:25561413 | ReViSP | Volume assessment of cancer spheroids |
| Software, algorithm | IVIS Spectrum In Vivo Imaging System | PerkinElmer | IVIS Spectrum In Vivo Imaging System RRID:SCR_018621 | Tumor volume analysis |
| Other | Lipofectamine RNAiMAX transfection reagent | Thermo Fisher Scientific | 13778075 | |
| Software, algorithm | ImageJ analysis software | NIH | ImageJ RRID:SCR_003070 | |
| Software, algorithm | TraceFinder v5.0 | Thermo Fisher Scientific | OPTON-30688 | |

