## [Decision Letter]

**Acceptance summary:**

Your work identifies DECR1 as an important regulatory component of polyunsaturated fatty acid oxidation in prostate cancer. Your evidence suggests that DECR1 expression contributes to prostate cancer progression and that blocking this enzyme may reverse metabolic processes that are critical for disease progression. Congratulations on your body of work!

**Decision letter after peer review:**

Thank you for submitting your article "DECR1 is an androgen-repressed survival factor that protects prostate tumor cells from ferroptosis" for consideration by *eLife*. Your article has been reviewed by three peer reviewers, and the evaluation has been overseen by a Reviewing Editor and Maureen Murphy as the Senior Editor. The reviewers have opted to remain anonymous.

The reviewers have discussed the reviews with one another and the Reviewing Editor has drafted this decision to help you prepare a revised submission.

Summary:

Fatty acid oxidation is important for some types of cancer. Inhibition of fatty acid oxidation has been explored as therapeutic strategy. Drugs that inhibit CPT1, the rate-limiting enzyme in fatty acid oxidation, are effective at reducing fatty acid oxidation but are associated with considerable toxicity. In this work, the authors identify DECR1, the rate-limiting enzyme for polyunsaturated fatty acid (PUFA) oxidation as a potential therapeutic target for prostate cancer. DECR1 expression is correlated with adverse tumor pathology and prognosis. DECR1 knockdown suppresses PUFA oxidation and cell growth / viability. The authors suggest that cell death induced by targeting DECR1 occurs through ferroptosis. Overall, this is a novel body of work that is interesting and well-written with significant merits. However, there are several items that should be addressed to improve the manuscript.

Essential revisions:

1) The conclusion that cells are dying via ferroptosis is insufficiently supported. Ferroptosis is characterized by the accumulation of phospholipid hydroperoxides which can be measured by mass spec or using the dye BODIPY C11. Are the PUFAs channeled into specific lipids (e.g. phospholipids) or do they remain as free fatty acids? Dose response curves should be performed in the presence and absence of multiple ferroptosis inducers to determine IC50s. What is the effect of deferoxamine on cell death? What are the effect of apoptosis inhibitors on the cells?

2) As it stands currently, the in vivo data are not convincing. In Figure 5 the authors stated that "LNCaP cells stably depleted of DECR1 showed highly variable growth rates in vivo, but inspection of the resultant tumors revealed significantly reduced cellular proliferation compared to control cells, concomitant with reduced DECR1 expression" the author should present the data for tumor growth and tumor volume. How do the authors explain the variability in tumor growth? For IHC, the authors should show a serial section for DECR1 and KI67 expression.

3) Some of the in vitro effects, e.g., cell death, are relatively modest. Are these effects intensified with stable knockdown or knockout approaches using shRNA or CRISPR/Cas?

---

## [Author Response]

Essential revisions:1) The conclusion that cells are dying via ferroptosis is insufficiently supported. Ferroptosis is characterized by the accumulation of phospholipid hydroperoxides which can be measured by mass spec or using the dye BODIPY C11. Are the PUFAs channeled into specific lipids (e.g. phospholipids) or do they remain as free fatty acids? Dose response curves should be performed in the presence and absence of multiple ferroptosis inducers to determine IC50s. What is the effect of deferoxamine on cell death? What are the effect of apoptosis inhibitors on the cells?

We have incorporated these points into our updated Figure 6 and new Figure 6—figure supplements 1 and 2 of the revised manuscript, and in the subsection “DECR1 targeting induces lipid peroxidation and cellular ferroptosis”. To summarise the changes:

In our original manuscript, we showed that DECR1 down-regulation led to accumulation of the free fatty acid linoleate (Figure 4C) and we now include new lipidomics data to show increased levels of PUFAs in the PC, π and PS classes of phospholipids (Figure 6A and B). PUFAs are highly susceptible to lipid peroxidation and ferroptosis and, in response to the reviewer’s questions, we have performed multiple additional experiments to further prove that DECR1 activity regulates ferroptosis.

Firstly, in the original manuscript we showed that DECR1 knockdown increased cellular levels of malondialdehyde, a marker of lipid peroxidation (Figure 6A, now Figure 6C). As suggested by the reviewer, we used the BODIPY C11 stain and intensity quantification, which revealed significant accumulation of phospholipid hydroperoxides, a hallmark of ferroptosis (new data Figure 6E). Our original data showed that DECR1 knockdown increases mitochondrial ROS levels (Figure 6B, now Figure 6D), while ectopic overexpression of DECR1 reduces mitochondrial ROS levels (Figure 6C, now Figure 6F). We extend these findings to show that ectopic overexpression of DECR1 also significantly reduces PUFA-induced ROS levels in the mitochondria (Figure 6G).

Secondly, DECR1 knockdown increased the cell sensitivity to three inducers of ferroptosis (erastin, FIN56 and ML210, compared to our original manuscript using only erastin); and we extended the dose ranges as requested to show that the three agents exhibited a lower IC_50_ in DECR1 downregulated cells than control cells (Figure 6J, K, Figure 6—figure supplement 1B). Moreover, we now show that two ferroptosis inhibitors, Ferrostatin-1 and Deferoxamine (as suggested), completely prevent the effects of DECR1 knockdown on LNCaP cell viability and cell death (Figure 6H and I).

Finally, we showed using flow cytometry that DECR1 knockdown-induced cell death is not characteristic of apoptosis (Figure 6—figure supplement 2A), and the apoptosis inhibitor ZVAD did not rescue the cells from cell death (Figure 6—figure supplement 2B). Collectively, these data (see revised Figure 6) strongly support ferroptosis as a primary mechanism of cell death in response to DECR1 targeting.

2) As it stands currently, the in vivo data are not convincing. In Figure 5 the authors stated that "LNCaP cells stably depleted of DECR1 showed highly variable growth rates in vivo, but inspection of the resultant tumors revealed significantly reduced cellular proliferation compared to control cells, concomitant with reduced DECR1 expression" the author should present the data for tumor growth and tumor volume. How do the authors explain the variability in tumor growth? For IHC, the authors should show a serial section for DECR1 and KI67 expression.

Subcutaneous growth of LNCaP tumor xenografts is widely acknowledged as being highly variable. Nevertheless, there was a significant decrease in proliferative index in the shDECR1 tumors compared with shControl cells. We have included a serial section for DECR1and Ki67 staining of representative tumors as requested (Figure 5—figure supplement 2B). We have also included the individual tumor growth curves as requested (Figure 5—figure supplement 2A).

In order to confirm these data as well as study the effect of DECR1 downregulation on PCa in the prostate microenvironment, we undertook a second study of orthotopically-grown LNCaP xenografts. These new data confirmed that DECR1 knockdown significantly inhibits tumor growth in the prostate (new data Figure 5K), but also significantly inhibits lung metastasis (new data Figure 5L), and have been added to the revised manuscript text:

“In vivo, LNCaP cells stably depleted of DECR1 showed highly variable growth rates in a subcutaneous model (Figure 5—figure supplement 2A), but inspection of the resultant tumors revealed significantly reduced cellular proliferation compared to control cells, concomitant with reduced DECR1 expression (Figure 5I, Figure 5—figure supplement 2B). To study the effect of DECR1 downregulation on PCa in the prostate microenvironment, we undertook a second study using LNCaP orthotopic xenografts. DECR1 knockdown significantly retarded tumor growth (Figure 5K, (Figure 5—figure supplement 2C, D, E), and significantly inhibited lung metastasis in the orthotopic tumor model (Figure 5L).”

3) Some of the in vitro effects, e.g., cell death, are relatively modest. Are these effects intensified with stable knockdown or knockout approaches using shRNA or CRISPR/Cas?

We used our shRNA stable knockdown LNCaP cells to assess this, showing a similar effect to the transient knockdown; with a ~25% decrease in cell viability and almost 2-fold increase in cell death detected (new data Figure 5B).